# High-Efficiency Removal of Cr(VI) from Wastewater by Mg-Loaded Biochars: Adsorption Process and Removal Mechanism

**DOI:** 10.3390/ma13040947

**Published:** 2020-02-20

**Authors:** Anyu Li, Hua Deng, Yanhong Jiang, Chenghui Ye

**Affiliations:** 1Key Laboratory of Ecology of Rare and Endangered Species and Environmental Protection, Ministry of Education, Guangxi Normal University, Guilin 541004, China; lianyu@stu.gxnu.edu.cn (A.L.); jiangjiangyh@sina.com (Y.J.);; 2Key Laboratory of Ecology of Rare and Endangered Species and Environmental Protection, Guangxi Normal University, Guilin 541004, China; 3School of Environment and Resources, Guangxi Normal University, Guilin 541004, China

**Keywords:** magnesium-loaded biochar, Cr(VI) removal, removal mechanism, wastewater, sorption

## Abstract

Biochars were produced with magnesium chloride as an additive for the sorption of hexavalent chromium dissolved in water using five types of straw (from taro, corn, cassava, Chinese fir, and banana) and one type of shell (*Camellia oleifera*) as the raw materials. The removal of hexavalent chromium by the six biochars mainly occurred within 60 min and then gradually stabilized. The kinetics of the adsorption process were second order, the Langmuir model was followed, and the adsorption of Cr(VI) by the six biochars was characterized by Langmuir monolayer chemisorption on a heterogeneous surface. Banana straw biochar (BSB) had the best performance, which perhaps benefitted from its special structure and best adsorption effect on Cr(VI), and the theoretical adsorption capacity was calculated as 125.00 mg/g. For the mechanism analysis, Mg-loaded biochars were characterized before and after adsorption by Fourier transform infrared spectroscopy (FTIR), X-ray diffractometry (XRD), and scanning electron microscopy/energy dispersive spectroscopy (SEM-EDS). The adsorption mechanism differed from the adsorption process of conventional magnetic biochar, and biochar interactions with Cr(VI) were controlled mainly by electrostatic attraction, complexation, and functional group bonding. In summary, the six Mg-loaded biochars exhibit great potential advantages in removing Cr(VI) from wastewater and have promising potential for practical use, especially BSB, which shows super-high adsorption performance.

## 1. Introduction

Chromium (Cr) waste is mainly generated by electroplating, leather making, chemical, pigment, metallurgy, refractory, and other industries [1], and in the aqueous environment, Cr mainly exists as Cr(VI) and Cr(III) [2]. Due to the high solubility, toxicity, mutagenicity, carcinogenicity, and teratogenicity of Cr(VI), it cannot be biodegraded and will accumulate in the food chain, thus causing damage to the human body [3]. Cr(VI) causes greater biotoxicity and environmental hazards than does Cr(III) [4]. Thus, the mass concentration of hexavalent Cr in wastewater must be strictly controlled and wastewater can only be discharged after reaching a standard.

At present, traditional methods for removing Cr from wastewater mainly include electrolysis [5], chemical methods [6], ion exchange methods [7,8], membrane separation [9], catalytic reduction [10], and adsorption [11,12]. The bioavailability, reactivity, and mobility levels of pollutants can be suitably controlled by adsorption. Therefore, it is an inevitable trend to study and prepare adsorbents with high adsorption capacities and low costs [13].

Biochar is produced via oxygen-limited pyrolysis from different biomass materials [14] that present carbon contents (achieved by controlling the pyrolysis temperature) as high as 51.60–85.02% [15] and are rich in acid-base groups, such as carboxyl (–COOH) and phenolic hydroxyl (–OH) groups [16]. Biochar is a new type of adsorbent with excellent adsorption properties for nutrient elements (ammonia nitrogen, phosphorus), organic dyes (Congo red, malachite green, etc.), and heavy metals (Cd, Cr, Cu, Pb, etc.) [16]. Therefore, researchers at home and abroad have studied the adsorption performance and mechanisms of biochar. For example, biochar modified with Fe_3_O_4_@SiO_2_-NH_2_ particles had a maximum adsorption capacity for hexavalent chromium ions of 27.20 mg/g, and its adsorption mechanism was composed of three steps for Cr(VI) on magnetic biochar [17]. A sulfuric acid pretreatment was applied, and then a MgO-coated biochar was used to prepare composite materials, which significantly improved the capacity for the adsorption of Cr(VI). A theoretical maximum adsorption capacity of 62.89 mg/g was estimated using the Langmuir model. The mechanism study showed that hexavalent chromium was adsorbed by the biochar through chemical interaction with MgO [18]. A new type of biochar-supported zero-valent iron nanocomposite (biochar-CMC-nZVI) was developed for the removal of hexavalent chromium from water. The adsorption capacity could reach 112.50 mg/g, thus proving that electrostatic attraction, reduction, and surface complexation are the main removal mechanisms. Moreover, this work showed the potential of biochar-CMC-nZVI as an efficient, green, and economical adsorbent for Cr(VI) [19]. A novel N-doped magnetic biochar was synthesized by the pyrolysis of an agar biomass loaded with ferric chloride. This char had a maximum adsorption capacity of 142.86 mg/g for hexavalent chromium [20].

In this work, the six starting materials (taro, corn, cassava, Chinese fir, banana, and *Camellia oleifera*) were loaded with a MgCl_2_ solution. The objectives were to (1) determine the Cr(VI) adsorption properties of the biochars obtained from the mentioned starting materials; (2) use various characterization techniques to determine the adsorption mechanisms; and (3) study the redox process of hexavalent chromium in adsorption onto nonmagnetic biochar. The adsorption mechanism was analyzed by a series of characterization methods, such as Fourier transform infrared spectroscopy (FTIR), X-ray diffractometry (XRD), scanning electron microscopy/energy dispersive spectroscopy (SEM-EDS), and XPS, which provided a scientific reference for understanding the effective mechanisms for wastewater treatment and the use of agricultural waste to prepare high-performance biochar. In addition, the pollution and damage of agricultural waste to the environment is very serious; thus, accelerating the process of resource utilization of agricultural waste has far-reaching significance for improving the ecological environment. Moreover, the use of agricultural waste to prepare high-performance adsorbent materials to treat wastewater is consistent with the principle of sustainable green chemistry.

## 2. Materials and Methods

### 2.1. Preparation of the Biochars

The preparation of biochars was achieved using a novel sample preparation method to improve the adsorption capacity of the adsorbent. The six raw biomass materials (taro, corn, cassava, Chinese fir, banana, and *Camellia oleifera*) all came from farmland near Guangxi Normal University, and the preparation process was the same for all materials. The materials were washed, pressed to eliminate excess water, and dried at 105 °C. Then, the materials were pulverized and passed through a 60-mesh sieve. Ten grams of 60-mesh biomass raw material was sieved, modified with 1 mol/L MgCl_2_, added to 150 mL MgCl_2_, shaken for 1 h, immersed at room temperature for 23 h, filtered, dried for 24 h, and carbonized in a tubular electric furnace at a temperature of 430 °C for 4 h. Biochars prepared from the six starting materials were labelled as BSB (from banana straw), CSB (from cassava straw), FSB (from Chinese fir straw), MSB (from corn straw), TSB (from taro straw), and CFSB (from *Camellia oleifera* shell biochar).

### 2.2. Characterization of the Biochars

The chemical compositions (carbon, hydrogen, nitrogen, and oxygen) of the six biochars were determined by an elemental analyzer (Eurovector, EA-3000, Gütersloh, Italy). The BET method was used for the determination of the surface area, total pore volume and average pore diameter by N_2_ adsorption (Micromeritics ASAP-2920, Norcross, GA, USA). An X-ray powder diffractometer (BRUKER D8 ADVANCE, Karlsruhe, Germany) was used to characterize the crystallization and chemical composition of the biochars before and after adsorption. SEM was used to observe the surface characteristics of the biochar samples in a field emission environment (Zeiss, EVO-18, Jena, Germany). A FTIR analyzer (Bruker Tensor 27, BRUKER AXS GMBH, Karlsruhe, Germany) was used to determine the functional groups in the biochars. The ultrastructure of the biochar was observed using high-resolution transmission electron microscopy (FEI Tecnai G2 F20, FEI, Portland, OR, USA).

### 2.3. Batch Adsorption Experiments

Samples of each prepared biochar (20 mg) were weighed to a conical flask, to which 50 mL of a 50 mg/L potassium dichromate (K_2_Cr_2_O_7_) solution was added, and the solution was then immediately placed in a constant-temperature oscillator to oscillate at 200 rpm. Samples were taken in conical flasks after 0, 10, 20, 40, 60, 90, 120, 150, 180, and 240 min, filtered with a 0.45 μm microporous membrane, and stored in 10 mL centrifuge tubes for sample determination. The determination of hexavalent chromium (Cr(VI)) was realized by the diphenylcarbazide spectrophotometry method. For the adsorption thermodynamics studies, samples of 20 mg of each prepared biochar were weighed into a series of conical flasks, and then 50 mL of 0.5, 1, 2, 4, 6, 8, 12, 24, 40, or 50 mg/L K_2_Cr_2_O_7_ solution was added. The pH of the adsorption solution was maintained constant, and the flask was removed after shaking at 20 °C, 30 °C, and 40 °C for 4 h at 200 rpm in a thermostatic oscillator. Cr(VI) was also determined by the diphenylcarbazide spectrophotometry method.

The adsorption capacities Q(mg/g) of the biochars were calculated by Equation (1):(1)$$Q=\left(C_{0}-C_{e}\right)\times V/m$$
where C_0_ is the initial concentration and C_e_ the equilibrium concentration, both in mg/L; V is the volume (mL), while m is the weight of the biochar (g).

### 2.4. Experiment Testing Regeneration Performance

The biochar desorption experiment was carried out using a 0.2 mol/L aqueous NaOH solution and a methanol solution as a desorbent. The adsorbed material (100 mg) was washed with a large amount of distilled water to remove unadsorbed Cr(VI). Then, the adsorbed biochar was vigorously stirred with 100 mL of 0.2 mol/L NaOH aqueous solution on a magnetic stirrer for 1 h. Finally, the biochar was vigorously stirred with 100 mL of methanol solution. The regenerated adsorbent was placed in a 50 mL, 50 mg/L Cr(VI) solution. After adsorption, the concentration of the adsorption solution was measured by the diphenylcarbazide spectrophotometry method, and adsorption and desorption of the biochar were carried out five times.

## 3. Results and Discussion

### 3.1. Physical Characteristics

Table 1 shows the elemental compositions of the biochars. The contents of the biochars prepared from different materials had notable differences, with FSB having the highest carbon content of 64.53%. To study and characterize the basic characteristics of the prepared Mg-loaded modified biochars, HRTEM, and zeta potential analyses were used to analyse the six biochars as shown in Figure 1. The HRTEM results show that the surfaces of the six biochars are covered with spherical or irregular particles, which may be compounds formed by supporting Mg. The particle size statistics were analyzed by Image J. The average particle diameters of BSB, FSB, MSB, CFSB, TSB, and CSB were 33.30, 6.44, 9.47, 3.27, 6.70, and 5.46 nm, respectively. The results indicate that the Mg-containing compounds were successfully loaded on the surface of the biochars, providing the possibility of ion exchange and other reactions (including complex precipitation, coordination, functional grouping, etc.). The surface of an adsorbent has a negative charge when pH > pH_PZC_ [21]. When the opposite is true, the surface is positively charged [22]. The values of pH and pH_PZC_ can be found in Table 1. The six biochars in the Cr(VI)-containing solution have pH < pH_PZC_. The surface of the biochar material is positively charged, which is beneficial to the adsorption of Cr(VI) in the solution (under natural pH, as shown in Table 1); therefore, the charge on the surface is positive. Dichromate (Cr_2_O_7_^2−^) in the solution can be electrostatically attracted [23] to biochar with a positive surface charge to promote the adsorption capability of Cr_2_O_7_^2−^. According to the complete set of mesoporous data, the surface areas and total pore volumes of the biochars before adsorption are smaller than those after adsorption, although the initial average aperture is greater than that after adsorption, thereby proving that biochar can effectively adsorb Cr_2_O_7_^2−^, which enters the pores of the biochar. We tested the contents of acid–base groups on the surfaces of the six biochars before and after adsorption by the Boehm titration method (Table 1). The contents of acid-base functional groups after adsorption all decreased. The comparison shows that the decrease in acid functional groups was greater than that in the basic functional groups, thus indicating that the contribution of the acid functional groups to the adsorption of Cr_2_O_7_^2−^ was greater than that of the basic functional groups [24].

### 3.2. Possible Mechanisms for Cr(VI) Adsorption onto the Biochars

#### 3.2.1. FTIR

The FTIR results before and after treatment with the Cr(VI) solutions for the six biochars are provided in Figure 2. All biochars showed similar trends in the spectrograms and had similar functional groups [16]: –OH (3429, 3430, 3427, 3431, and 3417 cm^−1^), –(OH)_2_–substituted aromatic rings or anthraquinones (1623, 1641, 1651, 1621, 1622, and 1630 cm^−1^), phenolic hydroxyl (1055, 1060, 1047, 1048, 1088, and 1039 cm^−1^), and phenols (610, 631, 622, 612, 614, and 608 cm^−1^). These groups are responsible for the binding of hexavalent chromium onto the six biochars, such as CSB [25]. The FTIR spectra before and after adsorption show important changes. The positions of –OH and –(OH)_2_–substituted anthraquinones or aromatic rings shift after adsorption and the peak widths’ decrease, which may provide evidence for hydrogen bonding during the process of adsorption [26]. Furthermore, the absorption peaks at 1039–1088 cm^−1^ and 608–631 cm^−1^ are not noticeable, suggesting that these groups present in biochar may be related to the Cr(VI) reduction [27].

#### 3.2.2. SEM/EDS

The SEM/EDS analysis was conducted before and after the adsorption of hexavalent chromium onto the biochars as shown in Figure 3. According to the SEM/EDS results of the six biochars before adsorption, the biochar itself has a suitable pore structure and a rough and convex surface, which is a prerequisite for adsorption [28]. Moreover, the figure shows that Mg ions in the six types of biochars almost covered the biochar surface, thus indicating the successful preparation of Mg-modified biochar. In addition, the elemental distribution map after adsorption shows that the biochar surface after reaction is covered with Mg, O, and Cr elements as background materials, thereby indicating that Mg/Cr compounds are uniformly deposited on the surface [24].

#### 3.2.3. XRD

The XRD patterns before and after the adsorption of Cr(VI) onto the six biochars, such as BSB, are shown in Figure 4. The biochar before adsorption was searched by the jade 5.0 phase and compared with the JCPDS card. The existence of Mg_2_(OH)_3_Cl·4H_2_O (PDF card 07-0412) is confirmed in BSB, CSB, and TSB at d = 8.17, 7.14, 5.99, 4.07, 3.87, and 2.46, thus indicating that Mg_2_(OH)_3_Cl·4H_2_O is the main magnesium-carrying phase of the three biochars. MgSO_3_·3H_2_O (PDF card 51-0530) exists in MSB at d = 6.70, 3.35, and 2.62, showing that MgSO_3_·3H_2_O is the main crystalline phase of the magnesium-charged MSB. Mg_4_Al (OH)_12_CO_3_·3H_2_O (PDF card 51-1528) exists in FSB at d = 7.61 and 3.78, which indicates that Mg_4_Al(OH)_12_CO_3_·3H_2_O is the main crystalline phase of the magnesium-charged FSB. MgO (PDF card 65-0476) exists in CFSB at d = 2.11 and 1.49, which indicates that the magnesium-charged CFSB has MgO as the main crystalline phase. The results of the magnesium treatment show that other compounds may be used to load the magnesium-modified biochar. According to the XRD patterns after adsorption, CFSB, CSB, TSB, FSB, and MSB mainly occur in the form of MgCrO_4_·7H_2_O (PDF card 01-0243), while BSB mainly occurs in the form of Cr(OH)_3_ (PDF card 12-0241). By combining these results with the kinetics results, it is shown that complexation precipitation is the main form of removing Cr from solutions [29].

### 3.3. Effect of the Solution pH

During the adsorption of Cr(VI) by biochar, the initial pH of the solution affects the surface charge distribution of the biochar, as shown in Figure 5. The six biochars follow similar laws. With increase in pH value, adsorption amount first exhibits a rising trend but then falls with continuing increase. This shows that pH value has a great influence on the adsorption performance of Cr(VI). The optimal pH value of the adsorption liquid is found to be 2, with the adsorption capacities of BSB, CSB, TSB, FSB, CFSB, and MSB being 112.26, 70.47, 45.78, 16.13, 18.15, and 38.38 mg/g, respectively. With a low pH solution (pH = 1–3), there is a large amount of H^+^ ions in the solution, which alters the surface charge, ionization degree, surface structure and ion morphology of the adsorption materials. Simultaneously, Cr(VI) mainly exists in the form of HCrO_4_^-^, which can combine with adsorbent to form a more stable compound that is beneficial to the adsorption of Cr(VI) by biochar. However, experiments show that adsorption capacity with a pH of 2 is higher than that with a pH of 1, which may be because the introduction of an excessive amount of H^+^ may destroy some functional groups on the biochar surface, leading to a decrease in adsorption capacity. Specifically, when the pH of the solution increases, functional groups on the surface of the biochar (such as –OH, –COOH, etc.) undergo proton dissociation to produce negatively charged functional groups. Furthermore, at high pH, HCrO_4_^−^ can also dissociate to form CrO_4_^2−^ and H^+^, which are not conducive to the adsorption of Cr(VI). Because the chromium reduction method is primarily used for chromium-containing industrial wastewater (especially electroplating wastewater), under acidic conditions (pH of approximately 2), the reducing agent (ferrous sulfate, sodium sulfite, etc.) is first added to Cr(VI) to obtain Cr(III), and then lime, sodium hydroxide, etc. are added to adjust the pH to form Cr(OH)_3_ precipitate. In this study, according to the actual wastewater treatment process, the pH of the solution was adjusted to approximately 2, and the functional groups contained in the biochar reduced some Cr(VI) to form a precipitate.

### 3.4. Adsorption Study

#### 3.4.1. Adsorption Kinetics Study

Adsorption kinetics are an important indicator of adsorption process design and operation control. They determine the adsorption rates of adsorbents for pollutants and ultimately determine the adsorption rates of adsorbents [30]. The adsorption fitting curves of the six biochars for Cr(VI) are shown in Figure 6. The prophase removal rate of Cr(VI) by TSB and CSB is faster than that of the other biochars. After adsorption for 20 min, the adsorption amount reaches 65.95% and 58.70% of the saturated adsorption amount, and with an increase in adsorption time, the adsorption amount continues to increase. The rising rates of TSB and CSB decrease after 90 min and 60 min, respectively, and the adsorption amounts reach 90.00% and 89.19% of the equilibrium adsorption amount, respectively, gradually reaching equilibrium. Additionally, the adsorption rates of TSB and CSB were calculated according to the pseudo-second-order kinetic equation and are 0.0995 mg/(g·min) and 0.4154 mg/(g·min), respectively, indicating that CSB has faster adsorption rates. BSB has the best adsorption capacity for Cr(VI). When the adsorption initially begins, the adsorption amount reaches 61.89 mg/g, with the process possibly benefitting from the large specific surface area of BSB (as shown in Table 1). FSB and CFSB have poor adsorption capacities for Cr(VI), reaching adsorption equilibrium at 150 min and 120 min, respectively. The adsorption of MSB is slow. After 5 min, the adsorption amount only reaches 9.97% of the saturated adsorption amount. Within 150 min, the adsorption amount increases steadily with increasing adsorption time, and adsorption equilibrium is reached at 180 min. Over the adsorption time, the adsorption sites on the surface of the biochar are gradually occupied by chromium ions, resulting in the occurrence of adsorption equilibrium. The curve in Figure 6a shows that the six biochars show a significant increase in the adsorption rate at the beginning of the adsorption and then slowly reach adsorption equilibrium, which is the phenomenon of “first fast and then slow” [15,31].

The adsorption mechanism was analyzed using pseudo-first-order equations, pseudo-second-order equations and the intraparticle diffusion model. The equations for the models are as follows [13,16]:

Pseudo-first-order kinetic equation:
(2)$$\ln\left(Q_{e}-Q_{t}\right)=\ln Q_{e}-K_{1}\times t$$

Pseudo-second-order kinetic equation:
(3)$${t/Q}_{t}{=1/K}_{2}Q_{e}{}^{2}{+t/Q}_{t},$$

Internal diffusion equation for particles:
(4)$$Q_{t}=K_{3}t^{1/2}+C,$$

Adsorption rate:
(5)$$h=K_{2}\times Q_{e}^{2}$$
where Q_t_ (mg/g) is the fraction of Cr(VI) adsorbed at time t; Q_e_ (mg/g) is the fraction of Cr adsorbed at equilibrium; K_1_ (1/min) is the pseudo-first-order rate constant; K_2_ (g/mg·min)) is the pseudo-second-order rate constant; and K_3_ (mg/(g·min^0.5^)) is the intraparticle diffusion rate constant. The values of these constants can be found by plotting ln(Q_e_−Q_t_) versus t and t/Q_t_ versus t; respectively. The constant C is related to the border effect. When the mechanism is intraparticle diffusion, Q_t_ and t^1/2^ are linearly related, and the straight line passes through the origin. h is the initial adsorption rate of the material, mg/(g·min)

The fitting parameters determined using pseudo-first-order kinetics and pseudo-second-order kinetics equations are shown in Table 2. The equilibrium concentration and experimental value calculated from the pseudo-first-order kinetics equation are known, the difference is large, and the correlation coefficient (R^2^) is relatively small, indicating that the pseudo-first-order kinetic equation is not suitable for describing the adsorption of Cr(VI) by the six biochars, such as BSB. The pseudo-second-order kinetics equation can well fit the adsorption process of Cr(VI) by the six biochars, and the fitted R^2^ is larger (all greater than 0.99), including the values for CFSB, FSB, TSB, MSB, CSB, and BSB. The theoretical adsorption amounts (19.53, 16.16, 47.37, 38.17, 73.53, and 111.11 mg/g, respectively) are close to the experimental values (20.24, 16.14, 31.80, 37.58, 68.34, and 110.20 mg/g, respectively), indicating that the adsorption of Cr(VI) by the six biochars in the experiment is more in line with the pseudo-second-order kinetics equation. At the same time, the pseudo-first-order kinetics equation is only suitable for describing the kinetics model of the initial stage of adsorption. The pseudo-second-order kinetics equation can describe all the stages of the adsorption reaction, including liquid film diffusion, surface adsorption, and internal diffusion. The diffusion process truly reflects the process of adsorption of Cr(VI) by biochar [32]. Based on the intraparticle diffusion model, it can be seen from Figure 6 that the six kinds of biochar fit two linear graphs, which indicates that the process of Cr(VI) adsorption is mainly divided into two steps. First, biochar was rapidly adsorbed on the surface. Subsequently, the adsorbate (Cr(VI)) diffused into the pores on the inner surface of the biochar. This pattern shows that the process is controlled by external mass transfer and then by intra-particle diffusion mass transfer [33]; when Cr(VI) diffuses into the adsorbent, the diffusion resistance gradually increases, resulting in a decrease in diffusion speed and eventually an adsorption equilibrium state.

#### 3.4.2. Adsorption Isotherm Study

The amount of biochar adsorbed increases as the adsorption concentration increases and finally reaches the adsorption equilibrium. The fitting was performed using the equations of the Freundlich and Langmuir models to elucidate the mechanism of the process and calculate the theoretical saturated adsorption capacity of biochar on Cr(VI).

Langmuir model:
(6)$$\mathrm{Qe}=\mathrm{CeQm}/\left(K_{L}+\mathrm{Ce}\right)$$

Freundlich model
(7)$${\mathrm{Qe}=K_{F}\mathrm{Ce}}^{1/n}$$
where Q_m_ (mg/g) is the maximum absorption capacity; Q_e_ (mg/g) is the fraction of Cr(VI) adsorbed; C_e_ (mg/L) is the equilibrium concentration of Cr(VI); C_0_ (mg/L) is the initial Cr(VI) concentration; K_L_ is the affinity constant; K_F_ is the Freundlich constant; and 1/n is the component factor (dimensionless).

The effectiveness of the adsorbent is expressed by the separation factor R_L_,
(8)$$R_{L}=1/\left(1+C_{0}K_{L}\right)$$

This equation shows how the initial concentration C_0_ influences the adsorption. Here, R_L_ = 0 is irreversible adsorption; R_L_ = 1 is linear adsorption; 0 < R_L_ <1 is effective adsorption; and R_L_ > 1 is disadvantageous adsorption.

The corresponding Gibbs free energy (ΔG^0^) can be determined using K_L_ calculated via the Langmuir model at different temperatures, and then the van’t Hoff equation is used to calculate the enthalpy changes (ΔH^0^) and entropy change (ΔS^0^) as
(9)$$\Delta G^{0}=-\mathrm{RT}\ln\left({\rho K}_{D}\right)$$
(10)$$\ln\left({\rho K}_{D}\right)\frac{\Delta S^{0}}{R}-\frac{\Delta H^{0}}{\mathrm{RT}}$$

In the formula, the Gibbs free energy units are kJ/mol, the units of the enthalpy change are J/(mol·K), and the units of entropy re kJ/mol; ρ is the density of water, 10^6^ mg/L; R is the gas constant, 8.314 J/(mol·K); T is the thermodynamic temperature, K.

The adsorption isotherms of the six biochars for Cr(VI) are shown in Figure 7. The different biochars displayed increasing amounts of adsorption as the concentration of Cr(VI) increased. At low concentrations, the adsorption amount of Cr(VI) increased almost linearly, and at high concentrations, the adsorption amount increased slowly and gradually became stable [19], achieving adsorption equilibrium. The appearance of this law can be explained by the fact that when the Cr(VI) solution is at a low concentration, the biochar can provide a large number of adsorption sites and acid-base functional groups (such as hydroxyl groups, phenolic hydroxyl groups, and aromatic hydrocarbons), thereby increasing the concentration of the adsorption solution and the adsorption capacity of biochar on Cr(VI). The adsorption capacities of the six types of biochars, such as CSB, increased with increasing adsorption temperature, indicating that increasing the adsorption temperature within a certain range is beneficial to the adsorption.

The experimental data were fitted using the Langmuir model and the Freundlich model. The adsorption isotherm fitting parameters are shown in Table 3. For the six biochars, the R^2^ values obtained by the Langmuir model at different temperatures were higher than the R^2^ values calculated by the Freundlich model and the theoretical saturated adsorption amount obtained by the Langmuir model was close to the experimental value. Therefore, the relationships of the six biochars with Cr(VI) in the experiment are more in line with the Langmuir model [34], indicating that the adsorption process is approximately the adsorption of monolayers. If the R_L_ value is in the range of 0–1, then the R_L_ values of all the biochars are less than 1, indicating that the adsorption of Cr(VI) by the biochars is beneficial adsorption [35,36]. The calculated thermodynamic parameters of adsorption are shown in Table 4. It can be seen from Table 4 that the ΔG^0^ values of the six biochars are all less than 0, indicating that adsorption occurs spontaneously. As the temperature increases, ΔG^0^ decreases, indicating that an increase in temperature is conducive to adsorption. The ΔH^0^ and ΔS^0^ values of the six biochars were all greater than 0, providing further evidence that the increase in temperature promoted adsorption. At the same time, we compared the adsorption capacity of six kinds of biochar without Mg loading. The theoretical saturated adsorption capacities of BSB, CFSB, TSB, CSB, FSB, and MSB were calculated by the Langmuir model to be 84.75, 12.92, 36.90, 47.17, 12.52, and 16.08 mg/g, respectively. A comparison of the Mg-modified biochar shows that the adsorption amount of Cr(VI) by the biochar after modification is remarkably improved (see Appendix A). Upon comparing the adsorption capacities for Cr(VI) of different biochars, the Cr(VI) adsorption amount achieved by BSB at 40 °C was the highest at up to 125.00 mg/g, indicating that BSB has great potential for use as a strong adsorbent. The experimental material removed phosphorus more efficiently than other tested materials (see Appendix A).

### 3.5. Cyclic Performance of Mg-Loaded Biochars

The recycling performance of the adsorbent is indispensable in the actual application process. The cyclic adsorption performance of the six modified biochars is studied by using 0.2 mol/L of a NaOH solution and methanol solution as the de-washing agent. As shown in Figure 8, the adsorption capacities of the six biochars TSB, CFSB, MSB, BSB, CSB, and FSB after five rounds of cyclic adsorption accounted for 76.79%, 64.41%, 81.07%, 80.00%, 75.25%, and 62.59% of the initial adsorption amount, respectively. BSB and MSB exhibited excellent recyclability. The evidence shows that biochar prepared from different types of magnesium biomass supported by magnesium chloride has stable adsorption performance in Cr(VI) solution.

## 4. Conclusions

In this work, six different Mg-loaded biochars were prepared to remove Cr(VI) from water. The prepared materials had magnesium-loaded compounds (Mg_2_(OH)_3_Cl·4H_2_O and MgO particles) and showed good capacities as adsorbents. BSB, CSB, TSB, FSB, CFSB, and MSB had the highest adsorption capacity at pH 2 at 114.64, 75.56, 50.21, 17.21, 20.21, and 40.12 mg/g, respectively. The adsorption thermodynamics of the six biochars could be explained by the Langmuir model and were pseudo-second order, and among them, BSB had an ultra-high adsorption capacity of up to 125.00 mg/g. Moreover, the FTIR, SEM/EDS, and XRD analyses indicated that the Cr adsorption occurred together with reduction, electrostatic attraction, functional group bonding, and complexation. Thus, Mg-loaded biochars could be considered applicable for the removal of Cr from wastewaters. In summary, this method of synthesizing Mg-loaded biochar offers new opportunities for finding effective and economical treatments to remove Cr(VI) and other heavy metal contaminants from wastewater.

## Figures and Tables

**Figure 1 materials-13-00947-f001:**
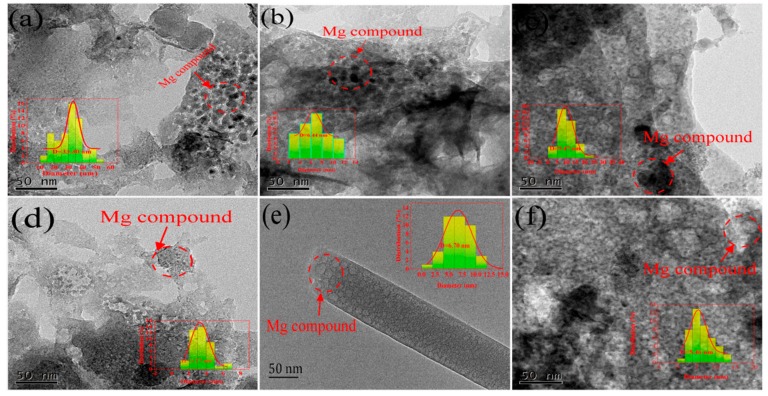
HRTEM images of six different Mg-loaded biochars. (**a**; **b**; **c**; **d**; **e**; and **f** represent BSB, FSB, MSB, CFSB, TSB, and CSB, respectively).

**Figure 2 materials-13-00947-f002:**
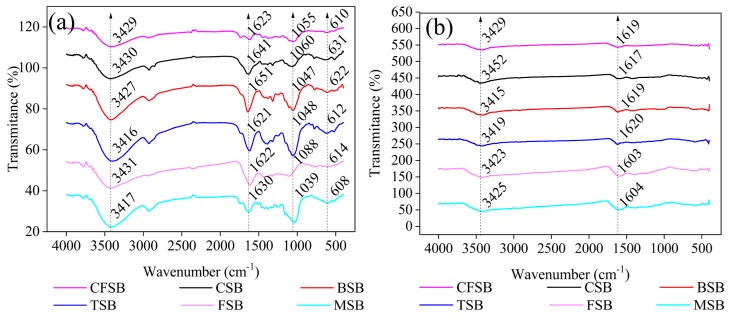
FTIR spectra of the six different biochars (**a**) before and (**b**) after adsorption of Cr(VI).

**Figure 3 materials-13-00947-f003:**
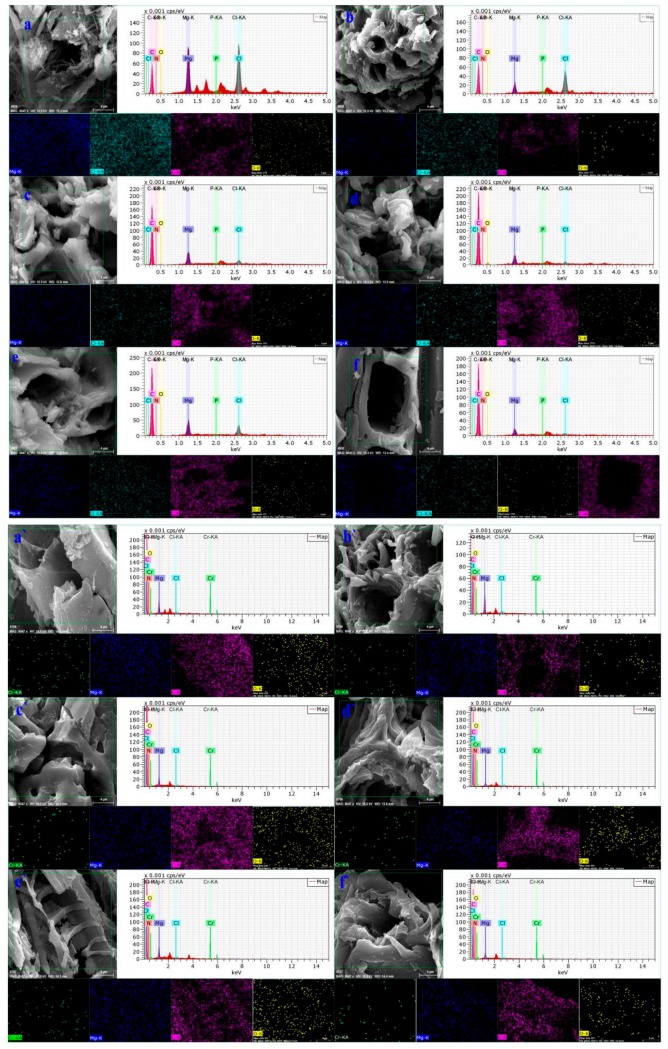
Scanning electron microscopic images of the six biochars. **a**, **a**’; **b**, **b**’; **c**, **c**’; **d**, **d**’; **e**, **e**’; and **f**, **f**’ represent before and after Cr(VI) adsorption on BSB, CSB, MSB, CFSB, TSB, and FSB, respectively.

**Figure 4 materials-13-00947-f004:**
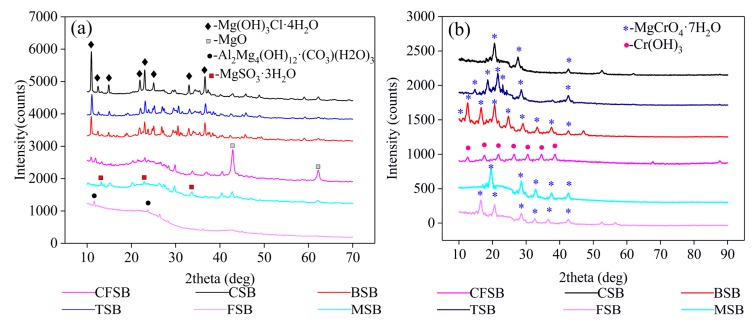
XRD analysis of the six biochars (**a**) before and (**b**) after adsorption of Cr(VI).

**Figure 5 materials-13-00947-f005:**
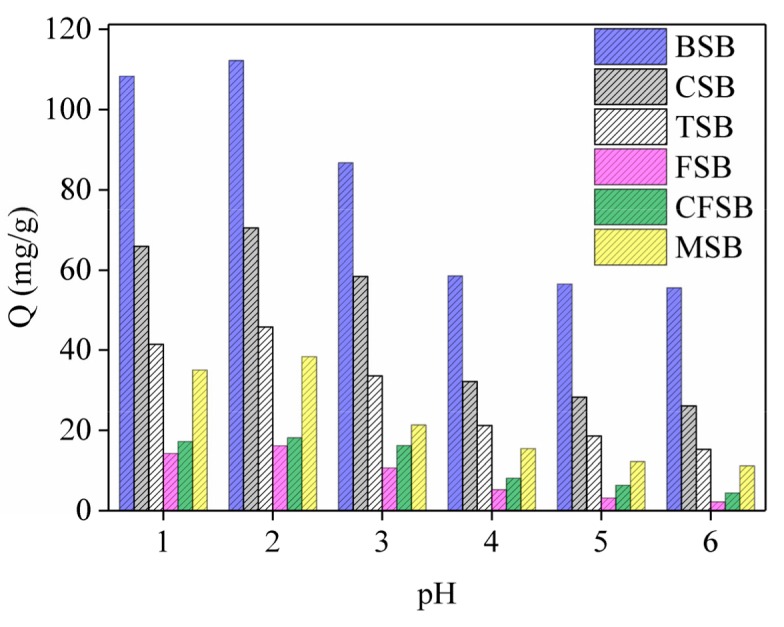
Effect of adsorption on Cr(VI) at different pH values.

**Figure 6 materials-13-00947-f006:**
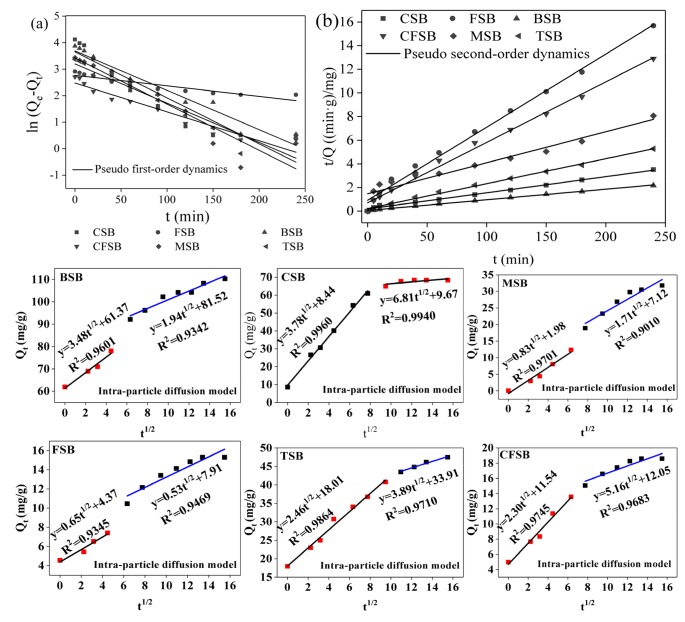
Adsorption kinetic curves of Cr(VI) on the six biochars ((**a**): Pseudo first-order dynamics fitting curve; (**b**): Pseudo second-order dynamics fitting curve).

**Figure 7 materials-13-00947-f007:**
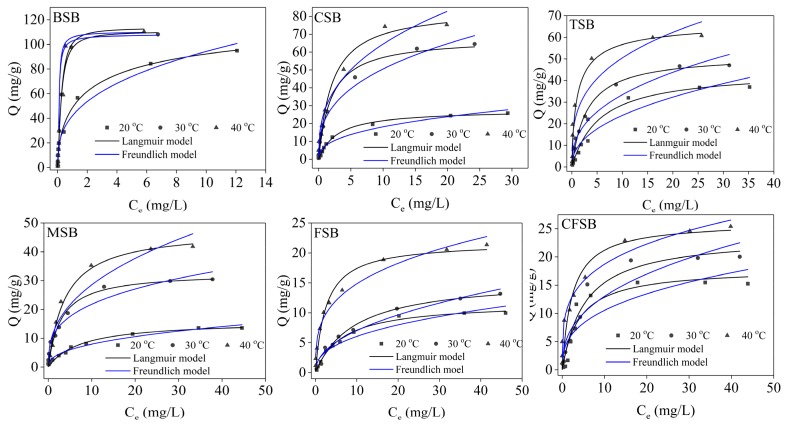
Adsorption isotherms of Cr(VI) for six different biochars.

**Figure 8 materials-13-00947-f008:**
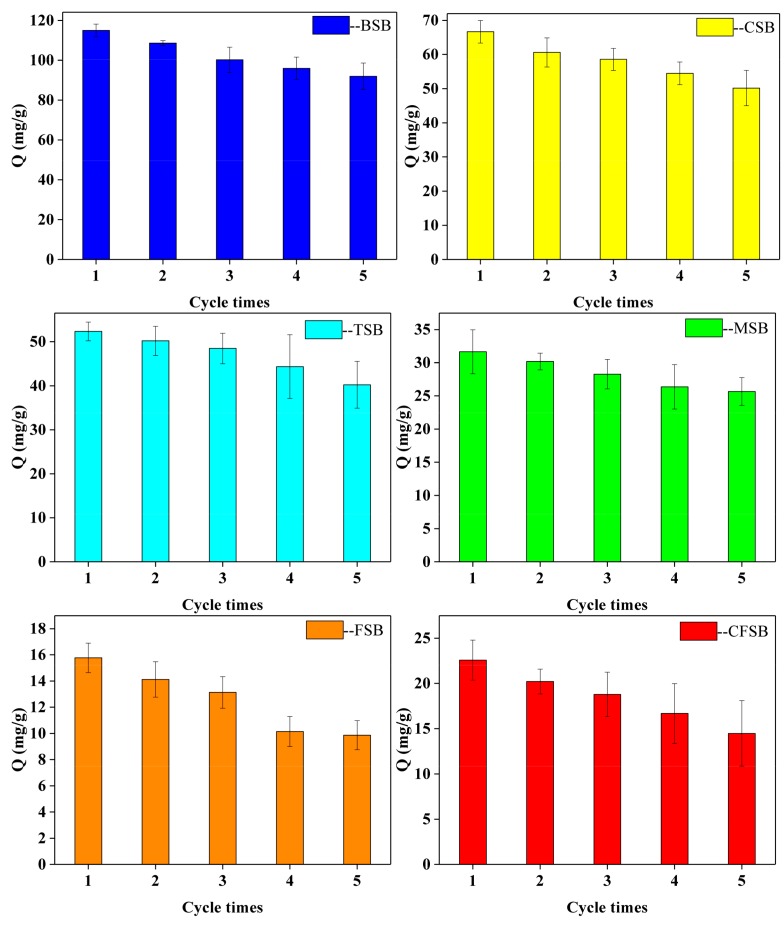
Recycle adsorption performance in wastewater of Cr(VI) adsorption by Mg-loaded biochars.

**Table 1 materials-13-00947-t001:** Basic characteristics of the biochars from six different raw materials.

Index		BSB	CSB	FSB	MSB	TSB	CFSB
pH		7.21	6.68	7.88	8.44	7.53	7.21
pH_PZC_ Biochar magnesium content (g/kg)		10.31	10.54	8.92	9.38	10.11	9.77
	before	190.62	113.21	75.09	65.95	181.95	98.20
BET (m^2^/g)	before	16.6839	10.3278	2.5272	9.3057	5.4411	4.3860
	after	36.3582	32.1380	27.6502	57.1085	33.6400	53.9261
Total pore volume (cm^3^/g)	before	0.0267	0.0406	0.0037	0.0263	0.0098	0.0114
	after	0.0900	0.0579	0.0408	0.0706	0.0446	0.0775
Average aperture (nm)	before	15.99	15.71	7.41	13.01	9.60	10.47
	after	9.91	7.22	5.91	4.95	5.31	5.76
Before adsorption	① (mmol/g)	1.1789	1.1181	0.7037	1.0300	1.1069	0.8729
	② (mmol/g)	1.1238	1.1216	1.1241	1.1168	1.1240	1.1192
After adsorption	① (mmol/g)	0.8523	0.8523	0.5193	0.6738	0.5688	0.4607
	② (mmol/g)	1.1189	1.0565	1.0325	0.7457	1.0565	1.0085
Mass composition	C (%)	34.66	47.13	64.53	48.98	44.73	34.73
	H (%)	4.81	7.81	3.31	2.96	1.86	1.86
	N (%)	0.85	0.66	0.14	1.42	1.69	1.69
	O (%)	20.31	22.41	26.05	23.48	21.22	20.89

Note: ①: represents acidic oxygen-containing functional groups; and ②: represents alkaline oxygen-containing functional groups.

**Table 2 materials-13-00947-t002:** Adsorption kinetics parameters for the six types of biochars.

Samples		Pseudo First-order Kinetics			Pseudo Second-order Kinetics			Intra-particle Diffusion						
		Q_e_ (mg/g)	K_1_ (1/min)	R^2^	Q_e_ (mg/g)	K_2_ (g/mg·min)	R^2^	C_1_ (mg/g)	K_d1_ (mg/(g·min^0.5^)	R^2^	C_2_ (mg/g)	K_d2_ (mg/(g·min^0.5^)	R^2^	Q_t_^①^ (mg/g)
**CFSB**	Cr(VI)	11.91	0.0109	0.9324	19.53	0.0038	0.9936	11.54	2.30	0.9745	12.05	5.16	0.9683	20.24
FSB	Cr(VI)	15.94	0.0040	0.8612	16.16	0.0042	0.9933	4.37	0.65	0.9345	7.91	0.53	0.9469	16.14
TSB	Cr(VI)	24.55	0.01148	0.9179	47.39	0.0021	0.9961	18.0.1	2.46	0.9864	33.91	3.89	0.9710	31.80
MSB	Cr(VI)	30.78	0.0174	0.8920	38.17	0.0015	0.9262	1.98	0.83	0.9701	7.12	1.71	0.9010	37.58
CSB	Cr(VI)	38.59	0.1730	0.8715	73.53	0.0055	0.9940	8.44	3.78	0.9960	9.67	6.81	0.9940	68.34
BSB	Cr(VI)	38.90	0.0148	0.9497	111.11	0.0015	0.9984	61.37	3.48	0.9601	81.52	1.94	0.9342	110.20

^①^ Actual amount of adsorption.

**Table 3 materials-13-00947-t003:** Adsorption parameters of the isotherm models for Cr(VI) for the six different biochars.

Samples			Langmuir				Freundlich		
T/K			Q_m_ (mg/g)	K_L_ (L/mg)	R^2^	R_L_	1/n	K_F_ (g/(mg·h))	R^2^
CSB	Cr(VI)	293	27.85	0.37	0.9910	0.0513–0.8438	0.5087	1.59	0.9415
		303	66.67	0.86	0.9908	0.0227–0.6993	0.4104	17.82	0.9786
		313	78.13	1.06	0.9836	0.0185–0.6535	0.3881	22.18	0.9343
FSB	Cr(VI)	293	11.68	0.08	0.9950	0.2000–0.9615	0.6482	1.43	0.9214
		303	15.77	0.11	0.9812	0.1538–0.9748	0.6204	1.61	0.9607
		313	22.03	0.50	0.9951	0.0385–0.8123	0.3967	6.12	0.9648
BSB	Cr(VI)	293	96.15	2.04	0.9936	0.0097–0.4950	0.5104	3.70	0.9276
		303	114.94	2.86	0.9955	0.0069–0.4115	0.4498	4.33	0.8635
		313	125.00	3.70	0.9394	0.0054–0.3509	0.3680	4.39	0.8021
CFSB	Cr(VI)	293	19.84	0.10	0.8465	0.1667–0.9524	0.7503	1.75	0.7869
		303	22.57	0.22	0.9854	0.0833–0.9009	0.5118	3.87	0.9382
		313	25.84	0.75	0.9935	0.0259–0.7220	0.2774	10.03	0.9539
MSB	Cr(VI)	293	15.22	0.18	0.9855	0.1000–0.9174	0.5084	2.54	0.9114
		303	31.65	0.58	0.9935	0.0333–0.7752	0.3972	9.04	0.9781
		313	47.18	0.68	0.9933	0.0286–0.7463	0.2026	20.61	0.9497
TSB	Cr(VI)	293	45.25	0.15	0.9798	0.1176–0.9302	0.6882	5.04	0.9352
		303	52.36	0.32	0.9986	0.0588–0.3472	0.6111	9.22	0.9192
		313	63.29	1.05	0.9960	0.0187–0.6557	0.4930	19.83	0.8086

**Table 4 materials-13-00947-t004:** Endothermic thermodynamic parameters of Cr(VI) by biochar.

Biochars	ΔG^0^ (KJ/mol)			ΔH^0^ (KJ/mol)	ΔS^0^ (J/(mol·K))
	293 K	303 K	313 K		
CSB	−31.23	−34.42	−36.10	40.28	244.86
FSB	−27.50	−29.24	−34.15	69.24	328.51
BSB	−35.39	−37.45	−39.36	22.90	199.04
CFSB	−28.05	−30.99	−35.20	76.79	357.12
MSB	−29.48	−33.43	−34.95	51.11	276.32
TSB	−29.03	−31.93	−36.08	73.75	350.13

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
