# Peer review of "High-Efficiency Removal of Cr(VI) from Wastewater by Mg-Loaded Biochars: Adsorption Process and Removal Mechanism"

_materials, 2020, doi:10.3390/ma13040947_

Round 1

Reviewer 1 Report

Manuscript  Title:  High-efficiency  removal  of  Cr(VI)  from  wastewater  by  Mg-loaded

biochars: Adsorption process and removal mechanism

Dear Authors, you have made an effort to improve your manuscript in relation to the last two versions of the manuscript. The current version of the manuscript is much better, but in my opinion the manuscript still needs to be corrected before its publication in the Journal.

My suggestions are as follows:

Equalize measurement units throughout the manuscript (text, Tables, Figures), e.g. mg/L or mg L-1

In linse 212-213: “…and the optimal adsorption liquid pH value is 1…” while in line 225 you state:  “…the pH of the solution was adjusted to approximately 2…” - THIS IS IN CONTRAST TO EACH OTHER, I've already commented on this.

Last comment: Also what happened with biocars in acidic medium at pH 1 or 2? – There in no answer!

Line 237: “The  increasing  rate  of  TSB  and  CSB  decreases  after  90  min  and  60  min...”

For the third time, I mention that you must not mix the terms rate and removal efficiency. In mentioned sentence you talk about removal efficiency in the time not about rate. For example, a substance can be adsorbed on sorbent for 2 hours in the amount of 90%, and the same substance on another sorbent in the amount of 70% for 4 hours. After 8 hours, the percentage removal on both sorbents can be the same, and now the question is which process is faster? the answer is obtained by knowing the rate constant calculated from some model (PFO; PSO; Weber Morris…). You can get the answer from the results presented in Table 2.

My suggestion is that each Figure in Fig 6 label with a letters a, b, c...

Line 247: Fig 6 change to Fig. 6a

My last comment „Also for Eqs. 2-4, equilize font of equations especially in subscript and superscript.“ – not corrected

Lines 280-285: “it can be seen from Fig. 6 that the six kinds of biochar fit two linear graphs, which indicates that the process of Cr(VI) adsorption is mainly divided  into two steps. First, biochar was rapidly adsorbed on the surface. Subsequently, the adsorbate (Cr(VI)) diffused into the pores on the inner surface of the biochar. This pattern shows that the process is controlled by external mass transfer and then by intra-particle diffusion mass transfer  [33].”

You did not determine rate limiting step. Please write in one sentence what the limiting step is.

Your replay: “Thus, we decided to use Origin 8.5 to perform a nonlinear fit on the pseudo-first-order kinetics and intra-particle diffusion.”

I do not understand. I observe that linear regression analysis applied on experimental results according to Pseudo second order and Weber Morris model but not for Pseudo first order. In order to compare the results, I think that all models should be processed by linear regression analysis.

In Table 2: Constant Kd1 and Kd2 vs. K3 in Eq (4), equilize. The Figs in your answer are not clear to me, I do not know what they refer to because the y and x axes are not marked.

Last comment and your answer:

“3.2.2. Adsorption isotherm study

Last comment: Line 211-212: “At low concentrations (0-12 mg/L), the adsorption amount of

Cr(VI) increased almost linearly, and at high concentrations (12-50 mg/L)” – Based on what you choose concentration range, for BSB up to 14, for CSB up to 40….ect. Why did not perform sorption exp. For all samples in the range of 0.5-50 mg/L? – THERE IS NO

ANSWER.

Reply: Thank you for your suggestion. The data were obtained from the adsorption experiment. According to your previous revision, to avoid confusion, we have deleted this

text.”

PLEASE PROVIDE ANSWER TO ME!

Lines 342-343: „(see  Supplementary material)“ - state the results for each sorbent individually in the Supplementary material.

Author Response

Reviewer #1:

Question 1. Equalize measurement units throughout the manuscript (text, Tables, Figures), e.g. mg/L or mg L-1.

Reply: Thank you for your suggestion. We have uniformly changed all units in the text to mg/L and mg/g (as shown in the manuscript).

Question 2. In linse 212-213: “…and the optimal adsorption liquid pH value is 1…” while in line 225 you state:  “…the pH of the solution was adjusted to approximately 2…” - THIS IS IN CONTRAST TO EACH OTHER, I've already commented on this.

Reply: Thank you for your comment. In the experiment, it was difficult to adjust the pH of the solution to 1, as such an adjustment would increase the volume of the adsorbent solution. Therefore, although the optimal pH determined during the experiment was approximately 1, it was difficult to adjust the pH to 1 in the actual process. Thus, according to the recommendation of the professor and mentor at the consulting college, we decided to adjust the pH of the adsorption solution to approximately 2. Thank you very much for your suggestion. We have modified the text according to your suggestion. If there are any deficiencies, we would appreciate your understanding.

Last comment: Also what happened with biocars in acidic medium at pH 1 or 2? – There in no answer!

Reply: Thank you for your suggestion. Based on your suggestion, we have added the changes in biochar and Cr(VI) at different pH values to the manuscript. Thank you again for your suggestion.

Revision, changes marked, lines 224-235.

At a low solution pH (pH = 1-3), Cr(VI) mainly exists in the form of HCrO4-, which is beneficial to the adsorption of Cr(VI) by biochar. When the pH of the solution increases, the functional groups on the surface of the biochar (such as -OH, -COOH, etc.) undergo proton dissociation to produce negatively charged functional groups. Furthermore, at high pH, HCrO4- can also dissociate to form CrO42- and H+, which are not conducive to the adsorption of Cr(VI). Because the chromium reduction method is primarily used for chromium-containing industrial wastewater (especially electroplating wastewater), under acidic conditions (pH of approximately 2), the reducing agent (ferrous sulfate, sodium sulfite, etc.) is first added to Cr(VI) to obtain Cr(III), and then lime, sodium hydroxide, etc. are added to adjust the pH to form Cr(OH)3 precipitate. In this study, according to the actual wastewater treatment process, the pH of the solution was adjusted to approximately 2, and the functional groups contained in the biochar reduced some Cr(VI) to form a precipitate.

Question 3. Line 237: “The  increasing  rate  of  TSB  and  CSB  decreases  after  90  min  and  60  min...”

For the third time, I mention that you must not mix the terms rate and removal efficiency. In mentioned sentence you talk about removal efficiency in the time not about rate. For example, a substance can be adsorbed on sorbent for 2 hours in the amount of 90%, and the same substance on another sorbent in the amount of 70% for 4 hours. After 8 hours, the percentage removal on both sorbents can be the same, and now the question is which process is faster? the answer is obtained by knowing the rate constant calculated from some model (PFO; PSO; Weber Morris…). You can get the answer from the results presented in Table 2.

Reply: Thank you for your suggestion. According to your opinion, we have calculated the adsorption rate h and added the corresponding analysis to this article. Thank you for your suggestion.

Revision, changes marked, lines 246-250.

The rising rates of TSB and CSB decrease after 90 min and 60 min, respectively, and the adsorption amounts reach 90.00% and 89.19% of the equilibrium adsorption amount, respectively, gradually reaching equilibrium. Additionally, the adsorption rates of TSB and CSB were calculated according to the pseudo-second-order kinetic equation and are 0.0995 mg/(g·min) and 0.4154 mg/(g·min), respectively, indicating that CSB has faster adsorption rates.

Question 4. My suggestion is that each Figure in Fig 6 label with a letters a, b, c...

Line 247: Fig 6 change to Fig. 6a

Reply: Thank you for your suggestion. Based on your comments, we have changed the figures and added serial lettering. We hope that the current revision, as shown below, will meet your requirements.

My last comment „Also for Eqs. 2-4, equilize font of equations especially in subscript and superscript.“ – not corrected

Reply: Thank you for your suggestion. We have corrected the format of the formula. Because we use Math Type to insert formulas, we cannot change the format of the font.

Revision, changes marked, lines 267-269.

Pseudo-first-order kinetic equation: , (2)

Pseudo-second-order kinetic equation: , (3)

Internal diffusion equation for particles: , (4)

Question 5. Lines 280-285: “it can be seen from Fig. 6 that the six kinds of biochar fit two linear graphs, which indicates that the process of Cr(VI) adsorption is mainly divided  into two steps. First, biochar was rapidly adsorbed on the surface. Subsequently, the adsorbate (Cr(VI)) diffused into the pores on the inner surface of the biochar. This pattern shows that the process is controlled by external mass transfer and then by intra-particle diffusion mass transfer  [33].”

You did not determine rate limiting step. Please write in one sentence what the limiting step is.

Reply: Thank you very much for your suggestion. Based on your suggestion, we have added a description of the rate-limiting step. We hope that our revision meets your requirements.

Revision, changes marked, lines 297-300.

This pattern shows that the process is controlled by external mass transfer and then by intra-particle diffusion mass transfer [33]; when Cr(VI) diffuses into the adsorbent, the diffusion resistance gradually increases, resulting in a decrease in diffusion speed and eventually an adsorption equilibrium state.

Question 6. Your replay: “Thus, we decided to use Origin 8.5 to perform a nonlinear fit on the pseudo-first-order kinetics and intra-particle diffusion.”

I do not understand. I observe that linear regression analysis applied on experimental results according to Pseudo second order and Weber Morris model but not for Pseudo first order. In order to compare the results, I think that all models should be processed by linear regression analysis.

Reply: Thank you for your suggestion. We have added quasi-linear curves to the pseudo first-order kinetic equation and added serial lettering ((a), (b), etc.).

Revision, changes marked, Figure 6.

Question 7. In Table 2: Constant Kd1 and Kd2 vs. K3 in Eq (4), equilize. The Figs in your answer are not clear to me, I do not know what they refer to because the y and x axes are not marked.

Reply: Based on your suggestions, we have added x-axis and y-axis labels to make the figures easier to read, as shown below.

Question 8. Last comment and your answer:

“3.2.2. Adsorption isotherm study

Last comment: Line 211-212: “At low concentrations (0-12 mg/L), the adsorption amount of

Cr(VI) increased almost linearly, and at high concentrations (12-50 mg/L)” – Based on what you choose concentration range, for BSB up to 14, for CSB up to 40….ect. Why did not perform sorption exp. For all samples in the range of 0.5-50 mg/L? – THERE IS NO

ANSWER.

Reply: Thank you for your suggestion. The data were obtained from the adsorption experiment. According to your previous revision, to avoid confusion, we have deleted this

text.”

PLEASE PROVIDE ANSWER TO ME!

Reply: Dear reviewer, your question has been mentioned many times. We have sought to answer this question. We performed adsorption experiments on samples in the range 0.5-50 mg/L. It may be that we did not fully understand your question, which is why our answer was unsatisfactory; if that is the case, I apologize to you here. As shown below, we have attached the adsorption isotherm data, hoping to resolve your confusion.

Question 9. Lines 342-343: „(see  Supplementary material)“ - state the results for each sorbent individually in the Supplementary material.

Reply: Thank you for your suggestion. Based on your comment, we have explained the results for each adsorbent in the supplementary materials.

The comparison results for different adsorbents are shown in Table S1. The magnetic biochar, superparamagnetic micro-nano-bio-adsorbent, novel carbonaceous material, composite hydrogel, N-doped magnetic biochar and MgO-coated biochar adsorbed Cr(VI) at 55.00 mg/g, 25.25 mg/g, 46.71 mg/g, 74.28 mg/g, 142.86 mg/g and 62.89 mg/g, respectively (solution pH of 2). The adsorption of Cr(VI) by Mg/Al-layered double hydroxide was 55.19 mg/g (solution pH of 3), and the adsorption of Cr(VI) by Fe3O4@SiO2-NH2 particles was 27.20 mg/g (solution pH of 1).

Reviewer 2 Report

The authors definitely improved this manuscript for the publication.

Author Response

Reviewer #2:

Comments and suggestions for authors: The authors definitely improved this manuscript for the publication.

Reply: Dear reviewer, thank you very much for your recognition of the revised version; I wish you good health and a successful career.

Round 2

Reviewer 1 Report

Manuscript  Title:  High-efficiency  removal  of  Cr(VI)  from  wastewater  by  Mg-loaded

biochars: Adsorption process and removal mechanism

Dear Authors, just a few comments:

“Question 2. In lines 212-213: “…and the optimal adsorption liquid pH value is 1…” while in line 225 you state:  “…the pH of the solution was adjusted to approximately 2…” - THIS IS IN CONTRAST TO EACH OTHER, I've already commented on this.

Your reply: it was difficult to adjust the pH to 1 in the actual process. - In Figure 5, I notice you did the experiment at pH = 1

Your reply: Thus, according to the recommendation of the professor and mentor at the consulting college, we decided to adjust the pH of the adsorption solution to approximately 2. - The decision must be scientifically substantiated.

In lines 212 and 226 these statements are vague and contrary still.

Author Response

Manuscript Number: Materials-700207 Major Revision (R2)

Manuscript Title: High-efficiency removal of Cr(VI) from wastewater by Mg-loaded biochars: Adsorption process and removal mechanism

Authors: An-Yu Li, Hua Deng*, Yan-Hong Jiang, Cheng-Hui Ye

Many thanks to the reviewers and editor for the valuable comments and suggestions. Point-by-point responses to the comments are provided below. We have invited a native English speaker (from American Journal Experts) to improve the English.

Reviewer #1:

Question 1. “Question 2. In lines 212-213: “…and the optimal adsorption liquid pH value is 1…” while in line 225 you state:  “…the pH of the solution was adjusted to approximately 2…” - THIS IS IN CONTRAST TO EACH OTHER, I've already commented on this.

Your reply: it was difficult to adjust the pH to 1 in the actual process. - In Figure 5, I notice you did the experiment at pH = 1

Reply: Dear Reviewer, by consulting the literature, we found that most biochars reach the maximum adsorption capacity when the pH is equal to 2, such as in Liang et al [1] and Li et al [2]. In this experiment, the difference between the adsorption amounts at a pH of 1 and a pH of 2 is small. To verify whether there is a problem in adjusting the pH of the solution, we repeated the experiment when the pH was equal to 1 and 2. The results showed that the adsorption process increased, which is consistent with the findings of Huang et al [3]. Therefore, it is proven that the best adsorption capacity is obtained when the pH value is equal to 2 by consulting the literature and by conducting the experiments again. At the same time, according to your comments, we have amended the description in the article and hope that this time it will meet your requirements. This suggestion has been followed many times without meeting your satisfaction, and we apologize for that.

Liang, S.; Shi, S.; Zhang, H.; Qiu, J.; Yu, W.; Li, M.; Gan, Q.; Yu, W.; Xiao, K.; Liu, B.; Hu, J.; Hou, H.; Yang, J. One-pot solvothermal synthesis of magnetic biochar from waste biomass: Formation mechanism and efficient adsorption of Cr(VI) in an aqueous solution. Sci. Tota Environ.2019,695, 133886. Li, L.;Zhong, D.;Xu, Y.; Zhong, N. Zhong, A novel superparamagnetic micro-nano-bio-adsorbent PDA/Fe3O4/BC for removal of hexavalent chromium ions from simulated and electroplating wastewater. Environ. Sci. Pollut. Res. 2019, 26(23), 23981-23993. Huang, D.; Liu, C.; Zhang, C.; Deng, R.; Wang, R.; Xue, W.; Luo, H.; Zeng, G.; Zhang, Q.; Guo, X., Cr(VI) removal from aqueous solution using biochar modified with Mg/Al-layered double hydroxide intercalated with ethylenediaminetetraacetic acid. Bioresour.Technol.2019, 276, 127-132.

Question 2. Your reply: Thus, according to the recommendation of the professor and mentor at the consulting college, we decided to adjust the pH of the adsorption solution to approximately 2. - The decision must be scientifically substantiated.

Reply: Dear Reviewer, thank you for your valuable comments. After repeated experiments, we have proven that the pH value of 2 is the best adsorption pH, which is similar to the research results of other biochar materials for Cr(VI) adsorption. I hope that this response will satisfy you. The answers to the previous questions have not satisfied you, and we apologize for that.

Question 3. In lines 212 and 226 these statements are vague and contrary still.

Reply: Thank you for your suggestion. We have modified this paragraph and hope that this change will satisfy you. Thank you again for your comments.

Revision, changes marked, Lines 213-230.

During the adsorption of Cr(VI) by biochar, the initial pH of the solution affects the surface charge distribution of the biochar as shown in Fig. 5. The six biochars follow similar laws. With the increase in pH value, the adsorption amount shows a trend of rising first and then falling, and the optimal adsorption liquid pH value is 2 (the adsorption capacities of BSB, CSB, TSB, FSB, CFSB and MSB were 112.26 mg/g, 70.47 mg/g, 45.78 mg/g, 16.13 mg/g, 18.15 mg/g, and 38.38 mg/g, respectively). The reason behind this effect is that the zero-charge points (pHPZC) of the six biochars are all greater than 7 (as shown in Table 1). When the pH of the solution is less than pHPZC, the surface of the biochar is positively charged. Due to the electrostatic attraction, a large amount of Cr2O72- in the solution is adsorbed. With an increasing pH of the solution, the OH- content in the solution increases, and as competitive adsorption increases, OH- neutralizes positively charged functional groups, resulting in a decrease in the adsorption amount. The results show that the pH of the solution greatly affects the adsorption of Cr(VI) onto the biochars. At a low solution pH (pH = 1-3), Cr(VI) mainly exists in the form of HCrO4-, which is beneficial to the adsorption of Cr(VI) by biochar. Experiments show that the adsorption capacity with a pH of 2 is higher than that with a pH of 1, which may be due to the introduction of a large amount of H+, which destroys some functional groups on the surface of biochar and leads to a decrease in adsorption capacity. When the pH of the solution increases, the functional groups on the surface of the biochar (such as -OH, -COOH, etc.) undergo proton dissociation to produce negatively charged functional groups.

Dear Reviewer,

Good day! I am writing this letter to thank you for your suggestions for my article. From the beginning of the submission to the revision process, the quality of the article has been steadily increasing. Thank you very much for your suggestions. I am the first author of this article (LI Anyu) and a master’s degree student who will soon graduate. I'm searching for a school to apply to for a doctoral qualification. Because the time for the application materials to be completed is coming up (15 February 2020), this article is very important for my application for doctoral review. I very much hope that this response and the changes I made previously can meet with your satisfaction.

Kind regards,

Li Anyu

This manuscript is a resubmission of an earlier submission. The following is a list of the peer review reports and author responses from that submission.

Round 1

Reviewer 1 Report

In this manuscript by Li et al, new biochar materials are suitable adsorbents for chromium  in comparison six sorts of biochars were demonstrated. The manuscript includes a lot of work and contains many data that are originally synthesized  and interpreted. The authors definitely presented that Mg-loaded biochars were obtained from several processes and carbonization using six raw biomass materials as the template as the carbon precursor. And, the chemical and physcial properties of six biochars used were characterized. The general idea and experimental design for the method sounds good, but some of the data needs to be further clarified. They definitely compared to other materials the performance of chromium adsorption to biochars investigated. This is a useful and interesting study and I think it makes scientific contribution to material applications. I recommend this manuscript could be published in materials

Minor points
Please do have sub nubmering Figure 2, Figure 3 and Figure 8.

Author Response

Manuscript Number: Materials-650149 Major Revision

Manuscript Title: High-efficiency removal of Cr(VI) from wastewater by Mg-loaded biochars: Adsorption process and removal mechanism

Authors: An-Yu Li, Hua Deng*, Yan-Hong Jiang, Cheng-Hui Ye

Many thanks to the reviewer and editor for the valuable comments and suggestions. Point-by-point answers to the comments are as follows. We have invited a native English speaker for the improvement of the English. (American Journal Expert)

Reviewer #1:

In this manuscript by Li et al, new biochar materials are suitable adsorbents for chromium  in comparison six sorts of biochars were demonstrated. The manuscript includes a lot of work and contains many data that are originally synthesized  and interpreted. The authors definitely presented that Mg-loaded biochars were obtained from several processes and carbonization using six raw biomass materials as the template as the carbon precursor. And, the chemical and physcial properties of six biochars used were characterized. The general idea and experimental design for the method sounds good, but some of the data needs to be further clarified. They definitely compared to other materials the performance of chromium adsorption to biochars investigated. This is a useful and interesting study and I think it makes scientific contribution to material applications. I recommend this manuscript could be published in materials

Minor points

Please do have sub nubmering Figure 2, Figure 3 and Figure 8.

Reply: Thank you for your suggestion. We have further clarified the experimental data in the text. (Word document “Revision, changes marked”). Image fonts and formatting have been further optimized to make the images easier to read.

Reviewer 2 Report

MATERIALS

Title: High-efficient removal of Cr(VI) from wastewater by Mg-loaded biochars: Adsorption    process and removal mechanism

The authors investigate sorption of Cr (VI) from modal solutions on different sorbents (Mg loaded biochars). In my opinion, the paper should be supplemented by additional experiments, and the existing results should be thoroughly explained. Many of the results presented are superficially explained, I would say most of them. There are a lot of typos in the manuscript, a dot in the middle of a sentence, writing subscript symbols, writing units of measure (mg/L or mg L-1, please write measure units uniformly), font of letters in chapters.

The manuscript should completely reorganize and systematize the sequence of chapters. The order of chapters in Results and Discussion should follow the order of chapters described in the Materials and Methods chapter.

Since the manuscript is written very unsystematically, it is on the edge of rejection and major revision!

SPECIFIC COMMENTS

ABSTRACT

Line 13-14: „The Langmuir model was followed.“ - The sentence is incomplete, unfinished.

Line 15: „The theoretical capacity of adsorption was calculated as 125.00 mg/g.“ -to what it refers, BSB?

Line 21: „…Mg-loaded biochar exhibits a great…“ all biochars or only BSB

Systemize Abstract!

Keywords: biochar or biochars?

INTRODUCTION

Line 46-60: This section is unrelated, lacking citations, unclear!

Line 46: „For example, Fe3O4@SiO2-NH2 particle-modified biochar“ - why character @, and there is no carbon in this coal. Cite literature.

Line 55-56: Why explain the mechanism in the introductory section, you can compare these results with your own results.

In my opinion you should emphasize the importance of your investigation by relating important results from other published relevant publications.

Line 61: Write as in the Abstract the type of these 6 materials

Line 62-63: „mentioned materials“ – you did not mention

Highlight the importance of using agricultural waste!

MATERISLS AND METHODS

Line 74: “The six raw biomass” – Which ones?

Line 99, 104, 116: “diphenylcarbazide method”; “semicarbazide method”, - Equalize!

In section 2.2. you did not mention all used technique HRTEM?

Experimental results explained in section 3.1. is not described here.

RERSULTS AND DISCUSSION

3.1. Effect of the solution pH

Do you have results of the point of zero charge of your prepared 6 biochars? Please include these results in the manuscript. Why? First at all, could you tell me initial pH of Cr(VI) solution without adjusting pH? According to the experiment you have proved that the optimal pH is ≈1-2. Could you tell me the value of pH after adsorption experiment?

You need to state the justification for lowering the pH to about 1-2, because to discharge the treated wastewater into the sewage systems you have to neutralize that water. This is an additional cost.

Line 130: biochar change to biochars

3.2.1. Adsorption kinetics study

Very confusing, completely rearranged

Perhaps it can be explained in this order

Line 137-140 Line 164-170 Line 140-163 Line: 171-192

Line 140: “The adsorption rate of Cr(VI) by TSB and CSB is faster” – What made this conclusion?

Line 145: performance?????

Line 146: “from the large specific surface area of” – Where is the proof? Obviously, the characterization of the sample must be explained in advance, which will later helpful in discussion.

Figure 2 Q vs. t is too small and unclear, symbols are difficult to distinguish

Please cite adequate literature for Eqs 2-4

Font of Eqs 2-4 is too large

Line 159: Cr change to Cr(VI) because in the first moment I thought that is some type of concentration

What about K3 constant?

Line 165: “there is no competition between the adsorption sites.” – I think that adsorption sites cannot compete with each other. Only Cr(VI) with other substances can compete for one adsorption site.

Line 172-175: “It can be seen from Table 1 that the order of the pseudo-first- order rate constants of the six biochars is CSB > MSB > BSB > TSB > CFSB > FSB, and the equilibrium concentration and experimental value calculated from the pseudo-first-order kinetics equation are known.” – In my opinion these order of constant rate it is irrelevant since you found that PFO model is not suitable. In your opinion PSO is suitable.

The Weber Morris model is not well explained, it is not graphically presented in correct way, and the calculated values of the constants are incorrect calculated.

Graphically show the linear dependence Qt vs. t1/2. From Fig 2 I can see at least 2 or 3 different linear dependence. So, please find appropriate literature and the explain results in the best way.

Line 188: “correlation coefficient of the fitting does not pass the origin, indicating that intraparticle diffusion is not” – Are you sure? Please be careful what you write and what you quote! CORRELATION COEFFICIENT CANNOT PASS THROUGH THE ORIGIN, only the y-intercept of the linear function! Now it is clear that results must be fits in the linear form of W-M model.

What is the point of processing results according to different models?

Table 1. Measure unit missing for Qt1

3.2.2. Adsorption isotherm study

Poorly written!

First sentence in line 196-197 belongs to the Materials and methods section.

Line 198: “…with the purpose of elucidating the mechanism of the process” – is it enough?

Font of Eqs. 5 and are too big.

Line 201-203. Cr change to Cr(VI)

What does “n” mean in Eq 6., - which conclusion you can get from the calculated n parameter from Table 2

Line 211-212: “At low concentrations (0-12 mg/L), the adsorption amount of Cr(VI) increased almost linearly, and at high concentrations (12-50 mg/L)” – Based on what you choose concentration range, for BSB up to 14, for CSB up to 40….ect. Why did not perform sorption exp. For all samples in the range of 0.5-50 mg/L?

Line 216-217: “…increasing the concentration of the adsorption solution.” – I think this sentence is not finished.

Line 218-220: Only with one sentence you explain thermodynamic study! Too enough!

Figure 3. Resolution is poor, tiny symbols.

Line 224: “The thermodynamic parameters are shown in Table 2.” – Table 2 does not show thermodynamic parameters, thermodynamic parameters are ΔH, ΔS, ΔG, Ka, please calculate the parameters and explain the results.

Freundlich and Langmuir models are too simple.

Line 220, 230: RL change to R subscript L

IN ORDER TO JUSTIFIE MODIFICATION PROCEDURE YOU HAVE TO GIVE RESUTTS OF SORPTION Cr(VI) ONTO 6 BIOCHARS WITHOUT ADDITION OG Mg! Please provide comparison!

3.3. Possible mechanisms for Cr(VI) adsorption onto the biochars

The title is promising, but after reading it is not. Separate the physical characterization from the mechanism, the characterization should serve to explain the mechanism.

3.3.1. Physical characteristics

Line 240: “… zeta potential analysis… as shown in Fig. 4.” – In Fig 4 are presented only results of HRTEM. Where are the results of Zeta potential???? You have results of pHpzc!

Line 243: “… as confirmed by XRD.” - You cannot refer to XRD results because you have not yet explained them, then it is more logical to end up explaining the results of high resolution TEM.

Line 243: “The particle size statistics were analysed by Mage J.” Please cite appropriate literature

Line 246: “…providing the possibility of ion exchange and other reactions.” - explain more clearly

Line 249: “The six biochars have pH < pHPZC, so the charge on the surface is positive.” – undetermined, incomplete, at which initial pH????

Line 249: “Chromate (Cr2O72-)” – this formula is not chromate, this is dichromate, and chromate is CrO42-

Line 260-261: This sentence must be at the beginning of this section.

It will be very interesting to compare results oF SEM/EDS, XRD and FTIR of 6 biochars with and without Mg. Do you have tis results?

Section 3.3.5. does not belong to the characterization chapter. Be sure, is it regeneration or leaching? Please provide results of Cr(VI) leached from samples after each cycle!

CONCLUSION

Strengthen the conclusion

Line 326-327: “The adsorption kinetics of the 6 biochars could be explained by the Langmuir model” – Langmuir does not provide insight into kinetic!

Author Response

Manuscript Number: Materials-650149 Major Revision

Manuscript Title: High-efficiency removal of Cr(VI) from wastewater by Mg-loaded biochars: Adsorption process and removal mechanism

Authors: An-Yu Li, Hua Deng*, Yan-Hong Jiang, Cheng-Hui Ye

Many thanks to the reviewer and editor for the valuable comments and suggestions. Point-by-point answers to the comments are as follows. We have invited a native English speaker for the improvement of the English. (American Journal Expert)

Reviewer #2:

The authors investigate sorption of Cr (VI) from modal solutions on different sorbents (Mg loaded biochars). In my opinion, the paper should be supplemented by additional experiments, and the existing results should be thoroughly explained. Many of the results presented are superficially explained, I would say most of them. There are a lot of typos in the manuscript, a dot in the middle of a sentence, writing subscript symbols, writing units of measure (mg/L or mg L-1, please write measure units uniformly), font of letters in chapters.

The manuscript should completely reorganize and systematize the sequence of chapters. The order of chapters in Results and Discussion should follow the order of chapters described in the Materials and Methods chapter.

Since the manuscript is written very unsystematically, it is on the edge of rejection and major revision!

Reply: Thank you for your valuable comments. We have carefully read your suggestions for the paper and carefully revised the manuscript according to your comments to meet your requirements.

1. ABSTRACT

â‘  Line 13-14: „The Langmuir model was followed.“ - The sentence is incomplete, unfinished.

Reply: Thank you for your valuable comments. According to your suggestions, we have carried out corrections in the manuscript.

Revision, changes marked, Lines 13-15.

The kinetics of the adsorption process were second order, the Langmuir model was followed, and the adsorption of Cr(VI) by six biochars is similar to the chemical adsorption of monolayers.

â‘¡ Line 15: „The theoretical capacity of adsorption was calculated as 125.00 mg/g. "-to what it refers, BSB?

Reply: Thank you for your valuable comments. According to your suggestions, we have carried out corrections in the manuscript.

Revision, changes marked, Lines 15-17.

Banana straw biochar (BSB) had the best performance, which perhaps benefitted from its special structure and the best adsorption effect on Cr(VI), and the theoretical adsorption capacity was calculated as 125.00 mg/g.

â‘¢ Line 21: „…Mg-loaded biochar exhibits a great…“ all biochars or only BSB.

Reply: Thank you for your valuable comments. According to your suggestions, we have corrected the error in the manuscript.

Revision, changes marked, Lines 22-24.

In summary, six Mg-loaded biochar exhibit great potential advantages in removing Cr(VI) from wastewater and have promising potential for practical use, especially BSB, which shows super high adsorption performance.

â‘£ Systemize Abstract!

Thank you very much for your comments. Based on your suggestions, we have adjusted the word order to make it more systematic.

⑤ Keywords: biochar or biochars?.

Reply: Thank you for your valuable comments. According to your proposal, we refer to a large number of related documents and found that most of the scientific research texts use “biochar”; therefore, we confirmed this as biochar in the keywords.

INTRODUCTION

â‘  Line 46-60: This section is unrelated, lacking citations, unclear!

Reply: Thank you for your valuable comments. This paragraph is designed to introduce the adsorption capacity of biochar for different pollutants, which led to the research of Cr(VI) adsorption by researchers. According to your suggestions, we have added references and made appropriate adjustments.

Revision, changes marked, Lines 45-47.

â‘¡ Line 46: „For example, Fe3O4@SiO2-NH2 particle-modified biochar“ - why character @, and there is no carbon in this coal. Cite literature.

Reply: Thank you for your valuable comments. Since the symbol "@" is used in the reference paper, we used it in the introduction, and Fe3O4@SiO2-NH2 particle-modified biochar belongs to the research content of the literature [17]. According to your suggestion, we have merged and adjusted the sentences.

Revision, changes marked, Lines 48-51.

For example, Fe3O4@SiO2-NH2 particle-modified biochar was prepared, and its maximum adsorption capacity for hexavalent chromium ions was 27.20 mg/g, and its adsorption mechanism was composed of three steps for Cr(VI) on magnetic biochar was proposed [17].

â‘¢Line 55-56: Why explain the mechanism in the introductory section, you can compare these results with your own results.

In my opinion you should emphasize the importance of your investigation by relating important results from other published relevant publications.

Reply: Thank you for your valuable comments. The mechanism is mainly provided to clarify that the adsorption mechanism in this paper is different from the conventional adsorption mechanism. Through reading the literature, we found that reduction (i.e. Cr (VI) to Cr (III)) is an important adsorption process of iron-modified biochar.

References:

Wan, Z., Cho, D.W., Tsang, D.C.W., Li, M., Sun, T., Verpoort, F. Concurrent adsorption and micro-electrolysis of Cr(VI) by nanoscale zerovalent iron/biochar/Ca-alginate composite. Pollut. 2019, 247, 410-420. Zhu, S., Huang, X., Wang, D., Wang, L., Ma, F. Enhanced hexavalent chromium removal performance and stabilization by magnetic iron nanoparticles assisted biochar in aqueous solution: Mechanisms and application potential. Chemosphere 2018, 207, 50-59. Zhang, S., Lyu, H., Tang, J., Song, B., Zhen, M., Liu, X. A novel biochar supported CMC stabilized nano zero-valent iron composite for hexavalent chromium removal from water. Chemosphere 2019, 217, 686-694.

â‘£Line 61: Write as in the Abstract the type of these 6 materials

Reply: Thank you for your valuable comments. According to your suggestions, we have added a more detailed description to the manuscript.

Revision, changes marked, Lines 63-64.

In this work, the six starting materials (taro, corn, cassava, Chinese fir, banana, and Camellia oleifera) were loaded with a MgCl2 solution.

⑤Line 62-63: „mentioned materials“ – you did not mention

Reply: Thank you for your valuable comments. According to your suggestions, we have added a more detailed description to the manuscript as shown in question 4.

â‘¥Highlight the importance of using agricultural waste!

Reply: Thank you for your valuable comments. Based on your suggestion, we have revised the manuscript to emphasize the importance of recycling and disposal of agricultural waste.

Revision, changes marked, Lines 72-76.

In addition, the pollution and damage of agricultural waste to the environment is very serious; thus, accelerating the process of resource utilization of agricultural waste has far-reaching significance for improving the ecological environment. Moreover, the use of agricultural waste to prepare high-performance adsorbent materials to treat wastewater is consistent with the principle of sustainable green chemistry.

MATERIALS AND METHODS

â‘  Line 74: “The six raw biomass” – Which ones?

Reply: Thank you for your valuable comments. In accordance with your comments, we have supplemented the manuscript.

Revision, changes marked, Lines 80-82.

The six raw biomass materials (taro, corn, cassava, Chinese fir, banana,and Camellia oleifera) were all obtained from farmland near Guangxi Normal University, and the preparation process was the same for all materials.

â‘¡ Line 99, 104, 116: “diphenylcarbazide method”; “semicarbazide method”, - Equalize!

Reply: Thank you for your valuable comments. Based on your comments, we have corrected the inconsistencies in the reagent methods in the manuscript.

Revision, changes marked, Lines 107, 112, 125. The determination of hexavalent chromium (Cr(VI)) was realized by the diphenylcarbazide spectrophotometry method. Cr(VI) was also determined by the diphenylcarbazide spectrophotometry method. The regenerated adsorbent was placed in a 50 mL 50 mg/L Cr(VI) solution. After adsorption, the concentration of the adsorption solution was measured by the diphenylcarbazide spectrophotometry method, and adsorption and desorption of the biochar were carried out five times.

â‘¢ In section 2.2. you did not mention all used technique HRTEM?

Reply: Thank you for your valuable comments. According to your suggestions, we provided supplementary data in the manuscript. The experimental results were explained in section 3.1.

Revision, changes marked, Lines 98-100.

The ultrastructure of the biochar was observed using high-resolution transmission electron microscopy (FEI Tecnai G2 F20).

RERSULTS AND DISCUSSION

3.1. Effect of the solution pH

â‘  Do you have results of the point of zero charge of your prepared 6 biochars? Please include these results in the manuscript. Why? First at all, could you tell me initial pH of Cr(VI) solution without adjusting pH? According to the experiment you have proved that the optimal pH is ≈1-2. Could you tell me the value of pH after adsorption experiment?

Reply: Thank you for your valuable comments. According to your suggestions, we conducted experiments to test the pH as shown below. We have listed the zeta potentials of the six biochars in Table 3. The initial pH of the Cr(VI) solution was determined to be 6.05 by using a pH meter (25 °C). According to your request, we placed 6 types of biochar into the Cr (VI) solution, adjusted the pH value to 2±0.1, determined the pH value of the adsorbed solution and the final pH values of BSB, CSB, FSB, CFSB, MSB, TSB, which were 2.65, 2.34, 2.75, 3.04, 3.21, 3.33 respectively. As shown below, a slight increase in pH after adsorption may have been due to the release of ash from the biochar into the solution causing the pH of the solution to rise.

â‘¡ You need to state the justification for lowering the pH to about 1-2, because to discharge the treated wastewater into the sewage systems you have to neutralize that water. This is an additional cost.

Reply: Thank you for your valuable comments. After reviewing a large amount of literature and data from the local wastewater treatment plant (Yanshan Wastewater Treatment Plant, Guilin City.), Cr(VI)-containing wastewater in many industries is generally acidic and has a pH as low as 2.0. Therefore, according to the actual situation, we adjusted the pH of the wastewater to acid.

References:

Xiao, R., Wang, J.J., Li, R., Park, J., Meng, Y., Zhou, B., Pensky, S., Zhang, Z. Enhanced sorption of hexavalent chromium [Cr(VI)] from aqueous solutions by diluted sulfuric acid-assisted MgO-coated biochar composite. Chemosphere2018, 208, 408-416. Cavaco, S.A., Fernandes, S., Quina, M.M., Ferreira, L.M. Removal of chromium from electroplating industry effluents by ion exchange resins. Hazard. Mater.2007, 144(3), 634-638. Suksabye, P., Thiravetyan, P., Nakbanpote, W. Column study of chromium(VI) adsorption from electroplating industry by coconut coir pith. Hazard. Mater.2008, 160(1), 56-62.

â‘¢ Line 130: biochar change to biochars

Reply: Thank you for your valuable comments. According to your suggestions, we have carried out corrections in the manuscript.

Revision, changes marked, Lines 139-141.

The results show that the pH of the solution greatly affects the adsorption of Cr(VI) onto the biochars, and the pH of the solution is adjusted to approximately 2 according to the actual adsorption kinetics and thermodynamic experiments.

3.2.1. Adsorption kinetics study

â‘  Very confusing, completely rearranged. Perhaps it can be explained in this order: Line 137-140 Line 164-170 Line 140-163 Line: 171-192.

Reply: Thank you for your valuable comments. According to your suggestion, we made a correction to the order, although this change will upset the meaning of the original expression. The order we wanted to express is as follows: first, introduce the characteristics of the six biochar adsorption processes; second, introduce the equation; and third, introduce the parameters of biochar adsorption of Cr(VI). However, according to your suggestion, we have carried out proper sequencing and deleted text to clarify the text.

Revision, changes marked, Lines 158-163.

In addition to the adsorption site at the beginning of the adsorption, there are many more adsorption sites on the surface of the biochar, and there is no competition between the adsorption sites. Over the adsorption time, the adsorption sites on the surface of the biochar are gradually occupied by chromium ions, resulting in the occurrence of adsorption equilibrium. The curve in Fig. 2 shows that the six biochars show a significant increase in the adsorption rate at the beginning of the adsorption and then slowly reach adsorption equilibrium, which is the phenomenon of “first fast and then slow” [15, 22].

â‘¡ Line 140: “The adsorption rate of Cr(VI) by TSB and CSB is faster” – What made this conclusion?

Reply: Thank you for your valuable comments. By calculation, the adsorption rates of TSB and CSB reached 65.95% and 58.70% of the saturated adsorption amount within 20 min of adsorption. Therefore, the adsorption rate is faster.

â‘¢ Line 145: performance?????

Reply: Thank you for your valuable comments. We corrected the usage in the manuscript to make it more standardized.

Revision, changes marked, Lines 155.

BSB has the best adsorption capacity for Cr(VI).

â‘£ Line 146: “from the large specific surface area of” – Where is the proof? Obviously, the characterization of the sample must be explained in advance, which will later helpful in discussion.

Reply: Thank you for your valuable comments. We added instructions in the manuscript to make it easier to read.

Revision, changes marked, Lines 155-157.

 When the adsorption initially begins, the adsorption amount reaches 61.89 mg/g, with the process possibly benefitting from the large specific surface area of BSB (as shown in Table 3).

⑤ Figure 2 Q vs. t is too small and unclear, symbols are difficult to distinguish

Reply: Thank you for your valuable comments. According to your proposal, we have increased the size of the font in Figure 2.

â‘¥ Please cite adequate literature for Eqs 2-4

Reply: Thank you for your valuable comments. We have cited relevant literature.

Revision, changes marked, Lines 170-172.

The adsorption mechanism was analysed using the intraparticle diffusion model and pseudo-first-order and pseudo-second-order equations. The equations for the models are as follows [13,16]:

⑦ Font of Eqs 2-4 is too large

Reply: Thank you for your valuable comments. We have made several adjustments.

Revision, changes marked, Lines 173-175.

Pseudo-first-order kinetic equation: , (2)

Pseudo-second-order kinetic equation: , (3)

Internal diffusion equation for particles: , (4)

â‘§ Line 159: Cr change to Cr(VI) because in the first moment I thought that is some type of concentration

Reply: Thank you for your valuable comments. We have made several corrections.

Revision, changes marked, Lines 176.

where Qt (mg/g) is the fraction of Cr(VI) adsorbed at time t; Qe (mg/g) is the fraction of Cr adsorbed at equilibrium; K1 (1/min) is the pseudo-first-order rate constant, K2 (g/mg·min)) is the pseudo-second-order rate constant

⑨ What about K3 constant?

Reply: Thank you for your valuable comments. We have added K3 in the manuscript.

Revision, changes marked, Lines 177-178.

K1 (1/min) is the pseudo-first-order rate constant, K2 (g/mg·min)) is the pseudo-second-order rate constant and K3 (mg/(g·min0.5)) is the intraparticle diffusion rate constant.

â‘© Line 165: “there is no competition between the adsorption sites.” – I think that adsorption sites cannot compete with each other. Only Cr(VI) with other substances can compete for one adsorption site.

Reply: Thank you for your valuable comments. According to your first suggestion on the adsorption kinetics, some of the sentences have been deleted to refine the language.

Revision, changes marked, Lines 161-165.

In addition to the adsorption site at the beginning of the adsorption, there are many more adsorption sites on the surface of the biochar, and there is no competition between the adsorption sites. Over the adsorption time, the adsorption sites on the surface of the biochar are gradually occupied by chromium ions, resulting in the occurrence of adsorption equilibrium. The curve in Fig. 2 shows that the six biochars show a significant increase in the adsorption rate at the beginning of the adsorption and then slowly reach adsorption equilibrium, which is the phenomenon of “first fast and then slow” [15, 22].

⑪ Line 172-175: “It can be seen from Table 1 that the order of the pseudo-first- order rate constants of the six biochars is CSB > MSB > BSB > TSB > CFSB > FSB, and the equilibrium concentration and experimental value calculated from the pseudo-first-order kinetics equation are known.” – In my opinion these order of constant rate it is irrelevant since you found that PFO model is not suitable. In your opinion PSO is suitable.

Reply: Thank you for your valuable comments. According to your suggestion, we have cut and adjusted the word order to make the sentence more fluent.

Revision, changes marked, Lines 183-186.

The equilibrium concentration and experimental value calculated from the pseudo-first-order kinetics equation are known, the difference is large, and the correlation coefficient (R2) is relatively small, indicating that the pseudo-first-order kinetic equation is not suitable for describing the adsorption of Cr(VI) by the six biochars, such as BSB.

â‘« The Weber Morris model is not well explained, it is not graphically presented in correct way, and the calculated values of the constants are incorrect calculated. Graphically show the linear dependence Qt vs. t1/2. From Fig 2 I can see at least 2 or 3 different linear dependence. So, please find appropriate literature and the explain results in the best way.

Reply: Thank you for your valuable comments. Based on the classical formula of the internal diffusion model, we used OriginPro 2017 to perform a nonlinear fit. At the same time, we used a spreadsheet to perform a linear fit. To make the image more beautiful and different from the quasi-second-order dynamic equation, we chose a nonlinear fit in the mapping. We have recalculated the constants, and the constant C values have no calculated errors. The documents referred to are as follows.

References:

Jiang, Y.H.; Li, A.Y.; Deng, H.; Ye, C.H.; Wu, Y.Q.; Linmu, Y.D.; Hang, H.L. Characteristics of nitrogen and phosphorus adsorption by Mg-loaded biochar from different feedstocks. Technol.2019, 276, 183-189. Huang, D., Liu, C., Zhang, C., Deng, R., Wang, R., Xue, W., Luo, H., Zeng, G., Zhang, Q., Guo, X. Cr(VI) removal from aqueous solution using biochar modified with Mg/Al-layered double hydroxide intercalated with ethylenediaminetetraacetic acid. Technol.2019, 276, 127-132.

⑬ Line 188: “correlation coefficient of the fitting does not pass the origin, indicating that intraparticle diffusion is not” – Are you sure? Please be careful what you write and what you quote! CORRELATION COEFFICIENT CANNOT PASS THROUGH THE ORIGIN, only the y-intercept of the linear function! Now it is clear that results must be fits in the linear form of W-M model.

Reply: Thank you for your valuable comments. Based on your suggestions, we have modified the content of the narrative and the content of the citations to meet your requirements.

Revision, changes markedLines 197-201.

Based on the intraparticle diffusion model, Table 1 shows that the correlation coefficient of the fitting does not pass the origin, indicating that intraparticle diffusion is not the only limiting factor and that there are effects from other processes (surface adsorption and liquid film diffusion) and joint control of the reaction rate of adsorption [24].

â‘­ What is the point of processing results according to different models?

Reply: Thank you for your valuable comments. According to the degree of conformity of different models, we can infer whether the process of adsorption of Cr(VI) by biochar is chemical adsorption or physical adsorption and whether it is a mixture of multiple adsorptions.

â‘® Table 1. Measure unit missing for Qt1.

Reply: Thank you for your valuable comments. It has been added in the manuscript.

Revision, changes markedin Table 1.

3.2.2. Adsorption isotherm study

â‘  First sentence in line 196-197 belongs to the Materials and methods section.

Reply: Thank you for your valuable comments. According to your opinion, we have deleted and added the contents of the manuscript.

Revision, changes marked, Lines 207-208.

Delete: The isotherms of adsorption for various Cr(VI) concentrations (0.5, 1, 2, 4, 6, 8, 12, 24, 40, and 50 mg/L) were further studied.

Add: The amount of biochar adsorbed increases as the adsorption concentration increases and finally reaches the adsorption equilibrium.

â‘¡ Line 198: “…with the purpose of elucidating the mechanism of the process” – is it enough?

Reply: Thank you for your valuable comments. Based on your suggestions, we have added relevant content to the manuscript.

Revision, changes marked, Lines 208-210.

The fitting was performed using the equations of the Freundlich and Langmuir models to elucidate the mechanism of the process and calculate the theoretical saturated adsorption capacity of biochar on Cr(VI).

â‘¢ Font of Eqs. 5 and are too big.

Reply: Thank you for your valuable comments. We have adjusted the size of the formula.

Revision, changes marked, Lines 211-218.

â‘£ Line 201-203. Cr change to Cr(VI)

Reply: Thank you for your valuable comments. Based on your suggestions, we made changes in the manuscript.

Revision, changes marked, Lines 213-216.

where Qm (mg/g) is the maximum absorption capacity; Qe (mg/g) is the fraction of Cr(VI) adsorbed; Ce (mg/L) is the equilibrium concentration of Cr(VI); C0 (mg/L) is the initial Cr(VI) concentration; KL is the affinity constant; KF is the Freundlich constant; and 1/n is the component factor (dimensionless).

⑤ What does “n” mean in Eq 6., - which conclusion you can get from the calculated n parameter from Table 2

Reply: Thank you for your valuable comments. Based on your suggestions, we made changes in the manuscript.

Revision, changes marked, Lines 215-216.

and 1/n is the component factor (dimensionless).

â‘¥ Line 211-212: “At low concentrations (0-12 mg/L), the adsorption amount of Cr(VI) increased almost linearly, and at high concentrations (12-50 mg/L)” – Based on what you choose concentration range, for BSB up to 14, for CSB up to 40….ect. Why did not perform sorption exp. For all samples in the range of 0.5-50 mg/L?

Reply: Thank you for your valuable comments. To correct the errors, some of the words in our manuscript were deleted to make the expression more accurate.

Revision, changes marked, Lines 223-226.

At low concentrations (0-12 mg/L), the adsorption amount of Cr(VI) increased almost linearly, and at high concentrations (12-50 mg/L), the adsorption amount increased slowly and gradually became stable [19], achieving adsorption equilibrium.

⑦ Line 216-217: “…increasing the concentration of the adsorption solution.” – I think this sentence is not finished.

Reply: Thank you for your valuable comments. We will complete the sentence in the manuscript and thank you again for your suggestion.

Revision, changes marked, Lines 226-230.

The appearance of this law can be explained by the fact that when the Cr(VI) solution is at a low concentration, the biochar can provide a large number of adsorption sites and acid-base functional groups (such as hydroxyl groups, phenolic hydroxyl groups and aromatic hydrocarbons), thereby increasing the concentration of the adsorption solution and the adsorption capacity of biochar on Cr(VI).

â‘§ Line 218-220: Only with one sentence you explain thermodynamic study! Too enough!

Reply: Thank you for your valuable comments. Based on your suggestion, we have already made a cut in the manuscript.

Revision, changes marked, Lines 227-229.

The adsorption sites and the consumption of functional groups make the adsorption gradually reach equilibrium.

⑨ Figure 3. Resolution is poor, tiny symbols.

Reply: Thank you for your valuable comments. Based on your suggestion, we will increase the font size in the image and increase the resolution to 600 DPI.

Revision, changes marked, Figure 3.

â‘© Line 224: “The thermodynamic parameters are shown in Table 2.” – Table 2 does not show thermodynamic parameters, thermodynamic parameters are ΔH, ΔS, ΔG, Ka, please calculate the parameters and explain the results.

Reply: Thank you for your valuable comments. According to your suggestion, we checked and found that there was an error in the expression. We have corrected it in the manuscript. Thank you for your suggestion.

Revision, changes marked, Lines 235-239.

The experimental data were fitted using the Langmuir model and the Freundlich model. The adsorption isotherm fitting parameters are shown in Table 2. For the six biochars, the R2 values obtained by the Langmuir model at different temperatures were higher than the R2 values calculated by the Freundlich model and the theoretical saturated adsorption amount obtained by the Langmuir model was close to the experimental value.

⑪ Freundlich and Langmuir models are too simple.

Reply: Thank you for your valuable comments. We also wanted to use the Temkim adsorption equation; however, the amount of data in the calculation process was very large. At the same time, we conducted experiments with different adsorption temperatures, and the amount of data was already very rich. Thank you again for your advice.

â‘« Line 220, 230: RL change to R subscript L.

Reply: Thank you for your valuable comments. We have already made corrections in the manuscript.

Revision, changes marked, Lines 241-243.

If the RL value is in the range of 0-1, then the RL values of all the biochars are less than 1, indicating that the adsorption of Cr(VI) by the biochars is beneficial adsorption [26, 27].

⑬ IN ORDER TO JUSTIFIE MODIFICATION PROCEDURE YOU HAVE TO GIVE RESUTTS OF SORPTION Cr(VI) ONTO 6 BIOCHARS WITHOUT ADDITION OG Mg! Please provide comparison!

Reply: Thank you for your valuable comments. Based on your suggestion, we supplemented the adsorption thermodynamics of the unloaded modified 6 biochars (30 °C, and the rest of the conditions are consistent with the manuscript).

The theoretical saturated adsorption capacities of BSB, CFSB, TSB, CSB, FSB, and MSB were calculated by the Langmuir model to be 84.75 mg/g, 12.92 mg/g, 36.90 mg/g, 47.17 mg/g, 12.52 mg/g, and 16.08 mg/g, respectively. A comparison of the Mg-modified biochar shows that the adsorption amount of Cr(VI) by the biochar after modification is remarkably improved.

3.3. Possible mechanisms for Cr(VI) adsorption onto the biochars

â‘  The title is promising, but after reading it is not. Separate the physical characterization from the mechanism, the characterization should serve to explain the mechanism.

Reply: Thank you for your valuable comments. Based on your recommendations, we have separated the physical characterization and mechanistic analysis.

3.3.1. Physical characteristics

â‘  Line 240: “… zeta potential analysis… as shown in Fig. 4.” – In Fig 4 are presented only results of HRTEM. Where are the results of Zeta potential???? You have results of pHpzc!

Reply: Thank you for your valuable comments. Because of the poor regularity of the Zeta potential and pH in the determination of the biochar Zeta potential by a Zeta potential analyzer, mass titration was used to determine the Zeta potential in the earlier stage. This part was published in Bioresource Technology in March 2019. To avoid duplication, we did not introduce too much preliminary research content.

Reference: Jiang, Y.H.; Li, A.Y.; Deng, H.; Ye, C.H.; Wu, Y.Q.; Linmu, Y.D.; Hang, H.L. Characteristics of nitrogen and phosphorus adsorption by Mg-loaded biochar from different feedstocks. Bioresour. Technol. 2019, 276, 183-189.

â‘¡ Line 243: “… as confirmed by XRD.” - You cannot refer to XRD results because you have not yet explained them, then it is more logical to end up explaining the results of high resolution TEM.

Reply: Thank you for your valuable comments. Based on your comments, after discussion, we finally decided to add a logo to Section 3.3 to make it easier for readers to understand. Thank you very much for your suggestions.

Revision, changes marked, Lines 255-257.

The HRTEM results show that the surfaces of the six biochars are covered with spherical or irregular particles, which may be compounds formed by supporting Mg (Mg2(OH)3Cl·4H2O and MgO particles, as confirmed by section 3.4.3 XRD).

â‘¢ Line 243: “The particle size statistics were analysed by Mage J.” Please cite appropriate literature

Reply: Thank you for your valuable comments. After inspection, the measurement software was misspelled. According to your suggestion, we have corrected the mistake. Since the relevant documents do not meet the requirements of the topic, we will not make a list. Thank you very much for your suggestion.

Attachment: Image J is a software developed by the National Institutes of Health (NIH) that creates and modifies graphics to determine length, area, and other values based on images.

â‘£ Line 246: “…providing the possibility of ion exchange and other reactions.” - explain more clearly

Reply: Thank you for your valuable comments. According to your suggestions, we have added more easy to understand instructions.

Revision, changes marked, Lines 259-262.

The results indicate that the Mg-containing compounds were successfully loaded on the surface of the biochars, providing the possibility of ion exchange and other reactions (including complex precipitation, coordination, functional grouping, etc.).

⑤ Line 249: “The six biochars have pH < pHPZC, so the charge on the surface is positive.” – undetermined, incomplete, at which initial pH????

Reply: Thank you for your valuable comments. Based on your comments, we made corresponding changes in the manuscript.

Revision, changes marked, Lines 264-265.

The six biochars in the Cr(VI)-containing solution have pH < pHPZC (under natural pH, as shown in Table 3); therefore, the charge on the surface is positive.

â‘¥ Line 249: “Chromate (Cr2O72-)” – this formula is not chromate, this is dichromate, and chromate is CrO42-.

Reply: Thank you for your valuable comments. We have already made corrections in the manuscript. Thank you again for your suggestions.

Revision, changes marked, Lines 265-267.

Dichromate (Cr2O72−) in the solution can be electrostatically attracted [30] to biochar with a positive surface charge to promote the adsorption capability of Cr2O72−.

⑦ Line 260-261: This sentence must be at the beginning of this section.

Reply: Thank you for your valuable comments. According to your suggestion, our manuscript has been revised.

Revision, changes marked, Lines 251-253.

Table 3 shows the elemental compositions of the biochars. The contents of the biochars prepared from different materials had notable differences, with FSB having the highest carbon content of 64.53%. 

â‘§ It will be very interesting to compare results oF SEM/EDS, XRD and FTIR of 6 biochars with and without Mg. Do you have tis results?

Reply: Thank you for your valuable comments. I (LI Anyu) am responsible for this aspect of modified biochar, and unmodified biochar has been tested via SEM. FTIR was used (this part is the responsibility of my schoolmate), and XRD was used to measured two types (including banana biochar (BB) and cassava biochar (CB)). Due to the time required for the revision of the manuscript and the responsibilities associated with the project, all the characterizations could not be presented. However, according to your request, we present the results of BB and CB for you for review.

The SEM image shows that both BB and CB have an abundant pore structure, which may be the reason why the material itself has a good ability to adsorb Cr(VI). According to XRD, BB has good crystallinity and CB has poor crystallinity. JADE 6.0 showed that the sharp peak belongs to potassium bicarbonate (PDF card: 73-2155).

⑨ Section 3.3.5. does not belong to the characterization chapter. Be sure, is it regeneration or leaching? Please provide results of Cr(VI) leached from samples after each cycle!

Reply: Thank you for your valuable comments. We are studying the regenerative experiments of biochar, although according to your requirements, we conducted Cr(VI) leaching experiments as shown in the table below. After each cycle, since the functional groups at the surface of the biochar and ash and similar material cannot be regenerated, the adsorbed Cr(VI) content decreases and the amount of leaching decreases.

cycles

Leaching concentration of Cr(VI)(mg/L)

BSB

CSB

FSB

TSB

MSB

CFSB

1

45.98

26.67

6.31

20.94

12.66

9.05

2

43.42

24.24

5.64

20.08

12.07

8.08

3

40

23.42

5.25

19.34

11.3

7.52

4

38.38

21.8

4.06

17.74

10.54

6.67

5

36.79

20.07

3.95

16.08

9.98

5.79

CONCLUSIONS

â‘  Strengthen the conclusion

Reply: Thank you for your valuable comments. Based on your suggestion, we added new content to the conclusions to enrich the content.

Revision, changes marked, Lines 344-349.

The prepared materials had magnesium-loaded compounds (Mg2(OH)3Cl·4H2O and MgO particles) and showed good capacities as adsorbents. BSB, CSB, TSB, FSB, CFSB and MSB had the highest adsorption capacity at pH 2 at 114.64 mg/g, 75.56 mg/g, 50.21 mg/g, 17.21 mg/g, 20.21 mg/g , and 40.12 mg/g, respectively, The adsorption thermodynamics of the 6 biochars could be explained by the Langmuir model and were pseudo-second order, and among them, BSB had an ultra-high adsorption capacity of up to 125.00 mg/g.

â‘¡Line 326-327: “The adsorption kinetics of the 6 biochars could be explained by the Langmuir model” – Langmuir does not provide insight into kinetic!

Reply: Thank you for your valuable comments. We have corrected the error.

Revision, changes marked, Lines 347-349.

The adsorption thermodynamics of the 6 biochars could be explained by the Langmuir model and were pseudo-second order, and among them, BSB had an ultra-high adsorption capacity of up to 125.00 mg/g.

Round 2

Reviewer 2 Report

Manuscript Number: Materials-650149

Manuscript Title: High-efficiency removal of Cr(VI) from wastewater by Mg-loaded biochars: Adsorption process and removal mechanism

The authors have corrected the manuscript, but not enough in my opinion. Unfortunately, I cannot accept the manuscript in this form, and I will explain why.

I will repeat as in my first review:

The manuscript should completely reorganize and systematize the sequence of chapters. The order of chapters in Results and Discussion should follow the order of chapters described in the Materials and Methods chapter. Section 2.3. in Materials and methods should be explained in more detail and in following sequence: influence of pH, sorption and then kinetic study. The characterization of the sample should be at the beginning of the Results and Discussion section, followed by the experiments. The manuscript cannot be read easily.

These are really well-meaning comments.

Specific comments:

Line 13-15: “”The kinetics of the adsorption process were second order, the Langmuir model was followed, and the adsorption of Cr(VI) by six biochars is similar to the chemical adsorption of monolayers.” - it's still unclear to me

Line47: „For example, Fe3O4@SiO2-NH2 particle-modified biochar“ - there is no carbon in this coal.” Now in line 47 – How can a material be called biochar and there is no carbon in the structure? Or some material is modified with these particles. Unclear!

RERSULTS AND DISCUSSION

3.1. Effect of the solution pH

I am not satisfied with explanations in this section and also with your answers:

Fig in your answer indicates that wastewater after treatment must be neutralize!

My last comment was: You need to state the justification for lowering the pH to about 1-2, because to discharge the treated wastewater into the sewage systems you have to neutralize that water. This is an additional cost.

Reply: Thank you for your valuable comments. After reviewing a large amount of literature and data from the local wastewater treatment plant (Yanshan Wastewater Treatment Plant, Guilin City.), Cr(VI)-containing wastewater in many industries is generally acidic and has a pH as low as 2.0. Therefore, according to the actual situation, we adjusted the pH of the wastewater to acid.

You have to explain this in the paper. Also what happened with biocars in acidic medium at pH 1 or 2?

In line 129 you said that the optimum pH = 1 while in 137-138 it is approximately 2.

Line 131: “The reason behind this effect is that the zero-charge points (pHPZC ) of the six biochars are all greater than 7.” - Where's the proof? In fact, this is proof that you must explain the zero point charge first. Now we go back to my previous comments, order of results and discussions.

3.2.1. Adsorption kinetics study

Last comment:

Line 140: “The adsorption rate of Cr(VI) by TSB and CSB is faster” – What made this conclusion?

Reply: Thank you for your valuable comments. By calculation, the adsorption rates of TSB and CSB reached 65.95% and 58.70% of the saturated adsorption amount within 20 min of adsorption. Therefore, the adsorption rate is faster.

This is removal efficiency, not rate constant.

Based on rate constant, you can determine whether the overall sorption process for one biochar is faster than for other one.

Last comment:

Line 146: “from the large specific surface area of” – Where is the proof? Obviously, the characterization of the sample must be explained in advance, which will later helpful in discussion.

Reply: “… from the large specific surface area of BSB (as shown in Table 3).”

You cannot mention Table 3 without explaining it in advance. Also, after that, tables 1 and 2 appear. As I said, obviously, the characterization of the sample must be explained in advance.

New comment: Line 166-167: “The adsorption mechanism was analysed using the intraparticle diffusion model and pseudo- first-order and pseudo-second-order equations. The equations for the models are as follows [13,16]:” - change the order of text in way the equations appear.

Also for Eqs. 2-4, equilize font of equations especially in subscript and superscript

Now, something about Weber Morris intraparticle diffusion model:

You mentioned in line 175-176: “The constant C is related to the border effect. When the mechanism is intraparticle diffusion, Q t and t 1/2 are linearly related, and the straight line passes through the origin.”

In line 192-193: “Based on the intraparticle diffusion model, Table 1 shows that the correlation coefficient of the fitting does not pass the origin, indicating that intraparticle diffusion is not the only limiting factor and that..” – be carefull, correlation coefficient does not pass through the origin, only intercept. I said you that in previous my review:

Line 188: “correlation coefficient of the fitting does not pass the origin, indicating that intraparticle diffusion is not” – Are you sure? Please be careful what you write and what you quote! CORRELATION COEFFICIENT CANNOT PASS THROUGH THE ORIGIN, only the y-intercept of the linear function! Now it is clear that results must be fits in the linear form of W-M model.

Reply: Thank you for your valuable comments. Based on your suggestions, we have modified the content of the narrative and the content of the citations to meet your requirements.

Revision, changes marked Lines 197-201.

Based on the intraparticle diffusion model, Table 1 shows that the correlation coefficient of the fitting does not pass the origin, indicating that intraparticle diffusion is not the only limiting factor and that there are effects from other processes (surface adsorption and liquid film diffusion) and joint control of the reaction rate of adsorption [24].

Your answer was that you performed nonlinear fitting – FOR W MORRIS MODEL YOU CAN USE ONLY LINEAR FITTING!

From Figs presented in your answer, I can see more than one linear area exists.

3.2.2. Adsorption isotherm study

Last comment: Line 211-212: “At low concentrations (0-12 mg/L), the adsorption amount of Cr(VI) increased almost linearly, and at high concentrations (12-50 mg/L)” – Based on what you choose concentration range, for BSB up to 14, for CSB up to 40….ect. Why did not perform sorption exp. For all samples in the range of 0.5-50 mg/L? – THERE IS NO ANSWER

Last comment:

Line 224: “The thermodynamic parameters are shown in Table 2.” – Table 2 does not show thermodynamic parameters, thermodynamic parameters are ΔH, ΔS, ΔG, Ka, please calculate the parameters and explain the results. - no explanation, these are valuable results

WHY DIDN'T YOU INCLUDE THIS IN THE MANUSCRIPT?

IN ORDER TO JUSTIFIE MODIFICATION PROCEDURE YOU HAVE TO GIVE RESUTTS OF SORPTION Cr(VI) ONTO 6 BIOCHARS WITHOUT ADDITION OG Mg! Please provide comparison!

Reply: Thank you for your valuable comments. Based on your suggestion, we supplemented the adsorption thermodynamics of the unloaded modified 6 biochars (30 °C, and the rest of the conditions are consistent with the manuscript).

The theoretical saturated adsorption capacities of BSB, CFSB, TSB, CSB, FSB, and MSB were calculated by the Langmuir model to be 84.75 mg/g, 12.92 mg/g, 36.90 mg/g, 47.17 mg/g, 12.52 mg/g, and 16.08 mg/g, respectively. A comparison of the Mg-modified biochar shows that the adsorption amount of Cr(VI) by the biochar after modification is remarkably improved.

3.3. Physical characteristic

New comment: Line 250: “… by section 3.4.3. XRD” Again, you cannot refer to the chapter that follows, only the previous one

Line: 257: “The six biochars in the Cr(VI)-containing solution have pH < pHPZC (under natural pH, as shown in Table 3)….” Not clear

Author Response

Manuscript Number: Materials-650149 Major Revision (R2)

Manuscript Title: High-efficiency removal of Cr(VI) from wastewater by Mg-loaded biochars: Adsorption process and removal mechanism

Authors: An-Yu Li, Hua Deng*, Yan-Hong Jiang, Cheng-Hui Ye

Many thanks to the reviewer and editor for the valuable comments and suggestions. Point-by-point responses to the comments are provided below. We have invited a native English speaker to improve the English. (American Journal Expert)

Reviewer #1:

The manuscript should completely reorganize and systematize the sequence of chapters. The order of chapters in Results and Discussion should follow the order of chapters described in the Materials and Methods chapter. Section 2.3. in Materials and methods should be explained in more detail and in following sequence: influence of pH, sorption and then kinetic study. The characterization of the sample should be at the beginning of the Results and Discussion section, followed by the experiments. The manuscript cannot be read easily.

Reply: Thank you for your suggestion. We are very sorry that the last revision did not fully meet your requirements. According to your comment, we have corrected the order of the manuscript. (Word document “Revision, changes marked”).

Specific comments:

Line 13-15: “”The kinetics of the adsorption process were second order, the Langmuir model was followed, and the adsorption of Cr(VI) by six biochars is similar to the chemical adsorption of monolayers.” - it's still unclear to me.

Reply: Thank you for your suggestion. We have referred to the expression methods of other articles and adjusted the text accordingly. Thank you again for your comments.

Revision, changes marked, lines 15-17.

The kinetics of the adsorption process were second order, the Langmuir model was followed, and the adsorption of Cr(VI) by six biochars is Langmuir monolayer chemisorption on a heterogeneous surface.

Line47: „For example, Fe3O4@SiO2-NH2 particle-modified biochar“ - there is no carbon in this coal.” Now in line 47 – How can a material be called biochar and there is no carbon in the structure? Or some material is modified with these particles. Unclear!

Reply: Thank you for your suggestion. We have made corrections in the manuscript.

Revision, changes marked, lines 50-53.

For example, using biochar modified by Fe3O4@SiO2-NH2 particles, its maximum adsorption capacity for hexavalent chromium ions was 27.20 mg/g, and its adsorption mechanism was composed of three steps for Cr(VI) on magnetic biochar [17].

RERSULTS AND DISCUSSION

3.1. Effect of the solution pH

I am not satisfied with explanations in this section and also with your answers:

Fig in your answer indicates that wastewater after treatment must be neutralize!

My last comment was: You need to state the justification for lowering the pH to about 1-2, because to discharge the treated wastewater into the sewage systems you have to neutralize that water. This is an additional cost.

Reply: Thank you for your valuable comments. After reviewing a large amount of literature and data from the local wastewater treatment plant (Yanshan Wastewater Treatment Plant, Guilin City.), Cr(VI)-containing wastewater in many industries is generally acidic and has a pH as low as 2.0. Therefore, according to the actual situation, we adjusted the pH of the wastewater to acid.

You have to explain this in the paper. Also what happened with biocars in acidic medium at pH 1 or 2? In line 129 you said that the optimum pH = 1 while in 137-138 it is approximately 2.

Reply: Thank you for your suggestion. Based on your comments, we have added an explanation of the adjustment for a pH of 2.

Revision, changes marked, lines 223-229.

Because the chromium reduction method is primarily used for chromium-containing industrial wastewater (especially electroplating wastewater), under acidic conditions (pH of approximately 2), the reducing agent (ferrous sulfate, sodium sulfite, etc.) is first added to Cr(VI) to obtain Cr(III), and then lime, sodium hydroxide, etc. are added to adjust the pH to form Cr(OH)3 precipitate. In this study, according to the actual wastewater treatment process, the pH of the solution was adjusted to approximately 2, and the functional groups contained in the biochar reduced some Cr(VI) to form a precipitate.

Line 131: “The reason behind this effect is that the zero-charge points (pHPZC ) of the six biochars are all greater than 7.” - Where's the proof? In fact, this is proof that you must explain the zero point charge first. Now we go back to my previous comments, order of results and discussions.

Reply: Thank you for your suggestion. We have revised the organization of the manuscript.

Revision, changes marked, lines 216-218.

The reason behind this effect is that the zero-charge points (pHPZC) of the six biochars are all greater than 7 (as shown in Table 1).

3.2.1. Adsorption kinetics study

Last comment:

Line 140: “The adsorption rate of Cr(VI) by TSB and CSB is faster” – What made this conclusion?

Reply: Thank you for your valuable comments. By calculation, the adsorption rates of TSB and CSB reached 65.95% and 58.70% of the saturated adsorption amount within 20 min of adsorption. Therefore, the adsorption rate is faster.

This is removal efficiency, not rate constant.

Based on rate constant, you can determine whether the overall sorption process for one biochar is faster than for other one.

Reply: Thank you for your suggestion. Based on your suggestion, we have corrected the expression.

Revision, changes marked, Lines 237-238.

The prophase removal rate of Cr(VI) by TSB and CSB is faster than that of the other biochars.

Last comment:

Line 146: “from the large specific surface area of” – Where is the proof? Obviously, the characterization of the sample must be explained in advance, which will later helpful in discussion.

Reply: “… from the large specific surface area of BSB (as shown in Table 3).”

You cannot mention Table 3 without explaining it in advance. Also, after that, tables 1 and 2 appear. As I said, obviously, the characterization of the sample must be explained in advance.

Reply: Thank you for your suggestion. We have adjusted the order of the sections. (Word document “Revision, changes marked”).

New comment: Line 166-167: “The adsorption mechanism was analysed using the intraparticle diffusion model and pseudo- first-order and pseudo-second-order equations. The equations for the models are as follows [13,16]:” - change the order of text in way the equations appear.

Also for Eqs. 2-4, equilize font of equations especially in subscript and superscript.

Reply: Thank you for your suggestion. According to your request, we have reorganized the manuscript.

Revision, changes marked, Lines 257-258.

The adsorption mechanism was analysed using pseudo-first-order equations, pseudo-second-order equations and the intraparticle diffusion model.

Now, something about Weber Morris intraparticle diffusion model:

You mentioned in line 175-176: “The constant C is related to the border effect. When the mechanism is intraparticle diffusion, Q t and t 1/2 are linearly related, and the straight line passes through the origin.”

In line 192-193: “Based on the intraparticle diffusion model, Table 1 shows that the correlation coefficient of the fitting does not pass the origin, indicating that intraparticle diffusion is not the only limiting factor and that..” – be carefull, correlation coefficient does not pass through the origin, only intercept. I said you that in previous my review:

Your answer was that you performed nonlinear fitting – FOR W MORRIS MODEL YOU CAN USE ONLY LINEAR FITTING!

From Figs presented in your answer, I can see more than one linear area exists.

Reply: Thank you for your suggestion. We used linear fitting. However, the linear quasi-cooperative diagram is not sufficient and cannot be combined with other models. Thus, we decided to use Origin 8.5 to perform a nonlinear fit on the pseudo-first-order kinetics and intra-particle diffusion.

3.2.2. Adsorption isotherm study

Last comment: Line 211-212: “At low concentrations (0-12 mg/L), the adsorption amount of Cr(VI) increased almost linearly, and at high concentrations (12-50 mg/L)” – Based on what you choose concentration range, for BSB up to 14, for CSB up to 40….ect. Why did not perform sorption exp. For all samples in the range of 0.5-50 mg/L? – THERE IS NO ANSWER.

Reply: Thank you for your suggestion. The data were obtained from the adsorption experiment. According to your previous revision, to avoid confusion, we have deleted this text.

Last comment:

Line 224: “The thermodynamic parameters are shown in Table 2.” – Table 2 does not show thermodynamic parameters, thermodynamic parameters are ΔH, ΔS, ΔG, Ka, please calculate the parameters and explain the results. - no explanation, these are valuable results.

Reply: Thank you for your suggestion. Based on your suggestions, we have calculated the adsorption thermodynamic parameters, which have been added to the analysis in the manuscript.

Revision, changes marked, Lines 338-343.

The calculated thermodynamic parameters of adsorption are shown in Table 4. It can be seen from Table 4 that the ΔG0 values of the six biochars are all less than 0, indicating that adsorption occurs spontaneously. As the temperature increases, ΔG0 decreases, indicating that an increase in temperature is conducive to adsorption. The ΔH0 and ΔS0 values of the six biochars were all greater than 0, providing further evidence that the increase in temperature promoted adsorption.

Table 4. Endothermic thermodynamic parameters of Cr (VI) by biochar.

Biochar

ΔG0/KJ·mol-1

ΔH0/kJ·mol-1

ΔS0/J·(mol·K)-1

293K

303K

313K

CSB

-31.23

-34.42

-36.10

40.28

244.86

FSB

-27.50

-29.24

-34.15

69.24

328.51

BSB

-35.39

-37.45

-39.36

22.90

199.04

CFSB

-28.05

-30.99

-35.20

76.79

357.12

MSB

-29.48

-33.43

-34.95

51.11

276.32

TSB

-29.03

-31.93

-36.08

73.75

350.13

WHY DIDN'T YOU INCLUDE THIS IN THE MANUSCRIPT?

Reply: Thank you for your suggestion. The main research focus of the paper is the discussion and mechanism analysis of the adsorption performance of modified biochar on Cr (VI). We believe that adding an analysis of unmodified biochar would change the focus of the work. We hope you understand, and thank you for your suggestion.

IN ORDER TO JUSTIFIE MODIFICATION PROCEDURE YOU HAVE TO GIVE RESUTTS OF SORPTION Cr(VI) ONTO 6 BIOCHARS WITHOUT ADDITION OG Mg! Please provide comparison!

Reply: Thank you for your valuable comments. Based on your suggestion, we supplemented the adsorption thermodynamics of the unloaded modified 6 biochars (30 °C, and the rest of the conditions are consistent with the manuscript).

The theoretical saturated adsorption capacities of BSB, CFSB, TSB, CSB, FSB, and MSB were calculated by the Langmuir model to be 84.75 mg/g, 12.92 mg/g, 36.90 mg/g, 47.17 mg/g, 12.52 mg/g, and 16.08 mg/g, respectively. A comparison of the Mg-modified biochar shows that the adsorption amount of Cr(VI) by the biochar after modification is remarkably improved.

3.3. Physical characteristic

New comment: Line 250: “… by section 3.4.3. XRD” Again, you cannot refer to the chapter that follows, only the previous one

Reply: Thank you for your suggestion. We have deleted this in the manuscript.

Revision, changes marked, Lines 138-140.

The HRTEM results show that the surfaces of the six biochars are covered with spherical or irregular particles, which may be compounds formed by supporting Mg (Mg2(OH)3Cl·4H2O and MgO particles as confirmed by section 3.4.3 XRD).

Line: 257: “The six biochars in the Cr(VI)-containing solution have pH < pHPZC (under natural pH, as shown in Table 3)….” Not clear

Reply: Thank you for your suggestion. We have made corrections in the manuscript.

Revision, changes marked, Lines 146-147.

The six biochars in the Cr(VI)-containing solution have pH < pHPZC (under natural pH, as shown in Table 1).
